# An Insight into Rational Drug Design: The Development of In-House Azole Compounds with Antimicrobial Activity

**DOI:** 10.3390/antibiotics13080763

**Published:** 2024-08-13

**Authors:** Daniel Ungureanu, Ovidiu Oniga, Cristina Moldovan, Ioana Ionuț, Gabriel Marc, Anca Stana, Raluca Pele, Mihaela Duma, Brîndușa Tiperciuc

**Affiliations:** 1Department of Pharmaceutical Chemistry, “Iuliu Hațieganu” University of Medicine and Pharmacy, 41 Victor Babeș Street, 400012 Cluj-Napoca, Romania; daniel.ungureanu@elearn.umfcluj.ro (D.U.); ooniga@umfcluj.ro (O.O.); cmoldovan@umfcluj.ro (C.M.); ionut.ioana@umfcluj.ro (I.I.); marc.gabriel@umfcluj.ro (G.M.); stana.anca@umfcluj.ro (A.S.); btiperciuc@umfcluj.ro (B.T.); 2“Prof. Dr. Ion Chiricuță” Oncology Institute, 34-36 Republicii Street, 400015 Cluj-Napoca, Romania; 3Department of Clinical Pharmacy, “Iuliu Hațieganu” University of Medicine and Pharmacy, 12 Ion Creangă Street, 400010 Cluj-Napoca, Romania; 4State Veterinary Laboratory for Animal Health and Safety, 1 Piața Mărăști Street, 400609 Cluj-Napoca, Romania; duma.mihaela-cj@ansvsa.ro

**Keywords:** azoles, organic synthesis, antimicrobial, heterocycles, structure-activity relationship

## Abstract

Antimicrobial resistance poses a major threat to global health as the number of efficient antimicrobials decreases and the number of resistant pathogens rises. Our research group has been actively involved in the design of novel antimicrobial drugs. The blueprints of these compounds were azolic heterocycles, particularly thiazole. Starting with oxadiazolines, our research group explored, one by one, the other five-membered heterocycles, developing more or less potent compounds. An overview of this research activity conducted by our research group allowed us to observe an evolution in the methodology used (from inhibition zone diameters to minimal inhibitory concentrations and antibiofilm potential determination) correlated with the design of azole compounds based on results obtained from molecular modeling. The purpose of this review is to present the development of in-house azole compounds with antimicrobial activity, designed over the years by this research group from the departments of Pharmaceutical and Therapeutical Chemistry in Cluj-Napoca.

## 1. Introduction

Antimicrobial resistance (AMR) is responsible for approximately 1.27 million direct deaths globally and has contributed to 4.95 million deaths, based on the latest statistics. The main cause of AMR includes the irrational use of antimicrobials in humans and animals, which lead bacteria such as *Acinetobacter baumanii* and *Pseudomonas aeruginosa* to acquire resistance against carbapenems; *Escherichia coli* against fluoroquinolones; *Klebsiella pneumoniae* against third-generation cephalosporins and carbapenems; *Staphylococcus aureus* against methicillin; and *Enterococcus faecium* against vancomycin [1,2,3,4]. 

AMR, which is a natural process that happens due to genetic mutations in pathologic bacteria to ensure their survival, is one of the top global threats to public health and development nowadays. AMR impacts countries across all regions and income levels, especially in low- and middle-income countries, and threatens many advances in modern medicine, making infections more difficult to treat and increasing the risk of surgical procedures or cancer chemotherapy. Deaths caused by resistant bacterial infections by 2050 will reach 10 million/year, more than the number of cancer-associated deaths, if no action is taken to mitigate this phenomenon [4,5,6,7].

Beyond causing death and disability, AMR imposes significant economic costs. The World Bank estimates that AMR could lead to an additional US$1 trillion in healthcare costs by 2050 and GDP losses ranging from US$1 trillion to US$3.4 trillion per year by 2030 [3].

AMR is a complex issue that demands both sector-specific actions in human health, food production, animal health, and environmental sectors, as well as a coordinated approach across these areas. The One Health approach is an integrated, unifying strategy aimed at achieving optimal and sustainable health outcomes for people, animals, and ecosystems. It acknowledges the close interconnection and interdependence of the health of humans, domestic and wild animals, plants, and the broader environment. The One Health approach to preventing and controlling AMR involves bringing together stakeholders from relevant sectors to collaborate in designing, implementing, and monitoring programs, policies, legislation, and research. The world is facing a crisis in both the development and accessibility of antibiotics, exacerbated by insufficient research in the field and constantly rising resistance levels. One of the key priorities in addressing AMR in human health is innovation and research, as well as the research and development of novel vaccines, diagnostics, and medicines. One of the key pillars of joint intersectoral and international initiatives to combat AMR is the development of new antibiotics that are effective against MDR pathogens. This is essential to prevent the onset of a post-antibiotic era [6,7,8,9,10,11].

Given that antibacterial therapy is typically short-term (from a few days to 1-2 weeks), combined with the very complex, expensive, and time-consuming drug development process and the rapid emergence of resistance, it takes a long time for the inventor company to see a return on their investment, severely reducing their interest in research in the field. Nowadays, pharmaceutical companies find more lucrative opportunities in developing treatments for chronic diseases such as diabetes, hypertension, mental illnesses, and cancer, where long-term use of medications ensures a steady and substantial revenue stream. Consequently, there has been a dramatic decrease in the development and approval of new antibiotics over the past four decades. Between 1980 and 1999, approximately 2.7 times more antibiotics were authorized compared to the period between 2000 and 2018 [8,12,13].

Considering the mentioned problems, the development of new antimicrobial agents is no longer the focus for large pharmaceutical companies; instead, it has become a research area for universities, research centers, start-ups, or small biotech companies [8,14].

At the Faculty of Pharmacy in Cluj-Napoca, Romania, within the departments of Organic Chemistry, Pharmaceutical Chemistry, and Therapeutical Chemistry, there has been a preoccupation for more than 50 years with the development of heterocyclic systems that contain azoles as central elements. This tradition was first founded by Professor Ioan Simiti and was kept over the years by his PhD students. As can be observed from the selective literature, the development of small bioactive molecules based on azole heterocycles is an ongoing trend and has implications for global concerns [15,16,17,18,19,20,21,22,23,24,25,26,27].

Azoles are a family of five-membered cyclic compounds that contain at least one nitrogen heteroatom and at least one non-carbon atom in the respective cycle. There are different subclasses generally named depending on the atomic arrangement, the number of nitrogen atoms present in the cycle, the substitution of the carbon atoms with exocyclic atoms, and the fusion with other aromatic or non-aromatic rings. More than half of all known organic compounds with biological activity are heterocycles. Statistically, 59% of the drugs approved by the Food and Drug Administration (FDA) contain nitrogen-based heterocycles [28]. 

Azoles that contain only the nitrogen heteroatom include pyrazole, imidazole, 1,2,3-triazole, 1,2,4-triazole, tetrazole, and pentazole. Between the antimicrobials containing the mentioned heterocycles, it is worth mentioning sulfaphenazole, ceftolozane, cefoselis (pyrazole derivatives), metronidazole, tinidazole, miconazole, ketoconazole, clotrimazole, albaconazole, econazole, terconazole, bifonazole, telithromycin (imidazole derivatives), fluconazole, itraconazole, voriconazole, isavuconazole, posaconazole, tazobactam, solithromycin, radezolid (triazole derivatives), and cefotetan, tedizolid, latamoxef, cefoperazone, cefamandole, and cefotiam (tetrazole derivatives) [29,30,31,32,33,34,35,36,37].

Other azoles that additionally contain the sulfur heteroatom are thiazole, isothiazole, 1,2,3-thiadiazole, 1,2,4-thiadiazole, 1,2,5-thiadiazole, and 1,3,4-thiadiazole. Between the antimicrobial agents containing at least a nitrogen atom together with a sulfur heteroatom, it is worth mentioning aztreonam, carumonam, pirazmonam, cefepime, cefpodoxime, cefoselis, ceftazidime, cefotiam, cefixime, ceftaroline, cefiderocol, sulfathiazole, ravuconazole, pramiconazole, isavuconazole, abafungin, myxothiazole (thiazole derivatives), and ceftolozane, ceftobiprole, ceftaroline, sulfamethizole (thiadiazole derivatives) [29,31,33,38,39,40,41].

Other azoles containing oxygen heteroatoms are oxazole, isoxazole, 1,2,3-oxadiazole, 1,2,4-oxadiazole, 1,2,5-oxadiazole, and 1,3,4-oxadiazole. Between the antimicrobial agents containing at least a nitrogen atom together with an oxygen heteroatom, it is worth mentioning flopristin, dalfopristin (oxazole derivatives) and sulfamethoxazole, sulfafurazole (sulfizoxazole), oxacilins, posizolid, contezolid, micafungin (isoxazole derivatives) [42,43,44].

The saturation of azoles can be variable, depending on the reagents used for their obtention. An increase in hydrogenation of rings would lead to a decrease in aromaticity and a decrease in rigidity, leading to flexibility of the rings. More than that, the hydrogenated or partially hydrogenated azoles can have supplementary exocyclic oxygen, sulphur, or nitrogen atoms linked by a double bond to the carbon atoms, transforming the respective heterocycles into -ones, -thiones, -imines or combinations of them if there are more than one type of exocyclic atom double bonded. This type of modification of the heterocycles leads to the appearance of much more complex and interesting physic-chemical and biological properties because the exocyclic atoms can act as hydrogen bond acceptors, favoring the interaction with biological targets, increasing the selectivity for a specific target, and reducing the affinity for the interfering targets. Moreover, the presence of exocyclic sulphur, oxygen, or nitrogen atoms in the immediate vicinity of the endocyclic nitrogen atoms leads to the appearance of tautomerism, rigidity of the respective region of the ring due to electron conjugation, or the appearance of the acidic properties of the N-H groups.

Between antimicrobials containing the mentioned heterocycles with carbon atoms substituted with double-bonded atoms, it is worth mentioning linezolid, tedizolid, posizolid, contezolid, radezolid, telithromycin, cethromycin, and solithromycin (oxazolidinone derivatives), penicillins (thiazolidine derivatives), pramiconazole, pirazmonam (imidazolidinone derivatives), itraconazole, and posaconazole (triazolone derivatives) [29,30,31,33,39,41,42].

Last but not least, some of the azoles are fused with the benzene ring to form condensed bicyclic azole derivatives, such as benzimidazole, indazole, benzotriazole, benzothiazole, and benzoxazole [28,45,46,47,48,49,50,51]. Between antimicrobials containing the azole fused to a benzene ring, is it worth mentioning caboxamycin and boxazomycin B (benzoxazole derivatives) [51,52].

Some of the currently FDA-approved azole drugs with antibacterial and antifungal activities are represented in Figure 1.

The complete characterization of the structure and knowledge about synthesis and physicochemical properties are very important for the advancement of the design of novel azole-containing drugs. Additionally, the azole-based derivatives can easily bind with the enzymes and receptors in organisms through noncovalent interactions such as hydrogen bonds, coordination bonds, ion-dipole, cation-π, π-π stacking, hydrophobic bonds, and van der Waals forces, thereby possessing various applications in medicinal chemistry [53,54].

Azole-based heterocycles are present in many natural products, biomolecules, and a large variety of drugs, including antitumor, natural antibiotics, anti-inflammatory, antidepressant, antimalarial, anti-HIV/antiviral, synthetic antibacterial, antidiabetic, herbicidal, fungicidal, and insecticidal agents. They have also been frequently used as a scaffold or central pharmacophore in synthetic human and animal pharmaceuticals and agrochemicals [53,54,55,56,57,58,59,60].

Among the azole heterocycles, thiazole represents a very important scaffold in medicinal chemistry. There are many compounds bearing this fragment that are currently used in therapy for the treatment of inflammation, oxidative stress, bacterial infections, hyperglycemia, hyperlipidemia, cancer, schizophrenia, hypertension, HIV infection, insomnia, allergies, etc. In new drug development studies, a combination of different pharmacophores in the same molecule may lead to novel compounds with higher or new biological activities [53,54,55,56,57,58,59,61,62,63,64,65].

In this regard and as a continuation of the tradition of Professor Simiti’s legacy, this review aimed to present the development of in-house azole compounds with antimicrobial potential, designed over the years by this research group from the departments of Pharmaceutical and Therapeutical Chemistry.

## 2. The Development of In-House Azole Compounds with Antimicrobial Activity

### 2.1. Synthesis and Antimicrobial Assay of Aryl and Hetaryl-1,3,4-Oxadiazoline Compounds

The oxadiazoline heterocycle is an important structural element that has been used for various chemical and biological purposes [66,67], such as the preparation of spirofused beta-lactam oxadiazolines or fused oxadiazepines for the treatment of Alzheimer’s disease [68]. In addition, oxadiazoline has been reported in compounds with different biological activities, which include antibacterial, antifungal, cytotoxic, and antitumor [69,70,71].

In order to synthesize oxadiazolines, various acyl hydrazones have been cyclized to 3-acyl-1,3,4-oxadiazolines under acylating conditions [66]. The acyl-hydrazone fragment represents another valuable scaffold to design biologically active compounds, supported by the pharmacophore potential of the -N=N-C(=O)- group. Compounds bearing an acyl-hydrazone moiety have been reported as potential antibacterial, antifungal, antiprotozoal, tuberculostatic, anticonvulsant, anti-inflammatory, and antitumor agents [72,73,74,75,76,77,78]. Moreover, these compounds can chelate metal ions, thus representing suitable ligands for the development of coordination compounds [78,79,80].

Starting from the literature studies that highlighted the importance of the oxadiazoline heterocycle in obtaining biologically active compounds, as well as the experience of this research group in the synthesis of heterocyclic structures, one of the main concerns was to obtain aryl and hetaryl-1,3,4-oxadiazoline hybrid compounds. The additional aromatic or heteroaromatic structure was usually grafted on the second position of the 1,3,4-oxadiazoline ring and was represented by 4-chloro-phenoxymethyl (**a**), pyridyl (**b**), 7-oxy-chromenyl (**c**), 2,4-bisthiazoles (**d**), 2-acetylamino-thiazole (**e**), and 2-aryl-thiazole (**f**) (Figure 2).

The precursors used in the synthesis of 1,3,4-oxadiazolines were the corresponding esters, previously converted into acyl-hydrazides by chemical treatment with excess hydrazine hydrate. These acyl-hydrazides were further derivatized to the corresponding acyl-hydrazones by condensation with various aromatic or heteroaromatic aldehydes in absolute ethanol. The concomitant acetylation and cyclization of acyl-hydrazones to 4-acetyl-4,5-dihydro-1,3,4-oxadiazol-2-yls were carried out by refluxing with acetic anhydride and sodium acetate (Figure 1, Figure 2, Figure 3 and Figure 4) [75,76,81,82,83].

The microbiological tests confirmed the great potential of this class of compounds as antibacterial agents against Gram-positive and Gram-negative strains, as well as antifungal agents [84].

Antimicrobial assays were performed on all the synthesized oxadiazolines against different bacterial strains and *Candida albicans* fungal strains using the agar diffusion method, with the evaluation of the inhibition zone (IZ) diameters [82,83].

The acyl-hydrazones (**1a**–**m**) and chromenyl-1,3,4-oxadiazoline (**2a**–**m**) compounds (Figure 3) were tested against *S. aureus*, *Bacillus subtilis*, *E. coli*, and *C. albicans*. However, no compound showed antibacterial or antifungal activity [82].

The replacement of the chroman-4-one structure with 2,4-bisthiazole (**3a**–**i**, Figure 4) proved to be beneficial, as these compounds showed weak antimicrobial activity against *B. subtilis* and *C. albicans* [83]. The compounds were inactive against *Staphylococcus* sp., *Streptococcus* sp., *Enterococcus* sp., *P. aeruginosa*, and *E. coli* [83]. Further replacement of the bisthiazole moiety with *N*-acetyl-thiazole in compounds **4a**–**i** (Figure 4) showed modest antibacterial activity against *B. subtilis*, *E. coli*, and *C. albicans* but inactivity against *S. aureus* [76,85].

The 2-pyridyl-thiazolyl-1,3,4-oxadiazolines (**6a**–**g**, Figure 5) and their corresponding acylhydrazones (**5a**–**g**, Figure 5) were tested against *S. aureus* ATCC 25923 (IZs = 2–20 mm), *E. coli* ATCC 25922 (IZs = 8–16 mm), *Salmonella typhimurium* ATCC 13311 (IZs = 10–16 mm), *Listeria monocytogenes* ATCC 35152 (IZs = 10–18 mm), *B. cereus* ATCC 13061 (IZs = 12–22 mm), and *C. albicans* ATCC 90028 (IZs = 10–22 mm) by measuring the growth inhibition zones in comparison to ciprofloxacin (IZs = 22–26 mm) and fluconazole (IZ = 28 mm). Based on these results, there was an increase in the overall antimicrobial activity by introducing the pyridyl ring as a substituent [86,87].

### 2.2. Synthesis and Antimicrobial Assay of Aryl and Hetaryl-1,3,4-Thiadiazoline Compounds

Overall, the oxadiazoline compounds showed modest antimicrobial activity or no activity at all against the tested bacteria and fungi. These observations led our research group to obtain some thiadiazoline-based compounds in which the oxygen heteroatom in oxadiazolines was replaced by a sulfur heteroatom.

Thiadiazolines and their acyclic thiosemicarbazones are important classes of nitrogen- and sulfur-containing compounds, being part of a large variety of compounds with promising biological implications and remarkable pharmacological properties, such as antibacterial, anthelmintic, antihypertensive, anti-inflammatory, analgesic, antioxidant, antiretroviral, and anticancer [72,88,89].

In recent decades, the synthesis of substituted thiadiazolines and related compounds has attracted considerable attention because of their importance as structural frameworks for pharmaceutical and industrial purposes [90]. Therefore, because of the promising pharmacophoric potential, our research group was interested in synthesizing compounds with this heterocycle linked to other heteroaromatic structures. The aromatic or heteroaromatic structures grafted on the 1,3,4-thiadiazoline ring in the fifth position were chromone moieties (**a**), substituted phenyl rings (**b**), and aryl-thiazoles (**c**), while in the second position of the 1,3,4-thiadiazoline rings, an *N*-acetyl group was grafted (Figure 6).

The general scheme of synthesis consisted of the cyclization of the corresponding thiosemicarbazones with acetic anhydride in pyridine. Simultaneously with the obtaining of the cyclic structure, the acetylation reaction took place on the nitrogen atom from the fourth position of the thiazolidine ring and on the amino group from the second position (Figure 6). The intermediate thiosemicarbazones were obtained under reflux by condensation of aromatic or heteroaromatic carbonyl compounds with various *N^4^*-substituted thiosemicarbazides. The reaction took place in absolute ethanol and in the presence of small amounts of concentrated sulfuric acid as a catalyst. The *N^4^*-substituted thiosemicarbazides were obtained from the reaction between hydrazine hydrate and various isothiocyanates at room temperature (rt) and by stirring in absolute ethanol (Figure 2, Figure 7) [91,92].

Antibacterial and antifungal assays were performed on *N*-(4-acetyl-5-aryl-4,5-dihydro-1,3,4-thiadiazol-2-yl)-acetamides (**8a**–**h**), *N*-(4-acetyl-5-(2-arylthiazol-4-yl)-4,5-dihydro-1,3,4-thiadiazol-2-yl) (**8i**–**j**), and their corresponding *N^1^*-arylidene-thiosemicarbazones (**7a**–**j**) (Figure 7). The compounds were screened for their antimicrobial activities against several bacterial strains and *C. albicans* ATCC 10231 by the agar diffusion method, using gentamycin (IZs = 8–22 mm) and fluconazole (IZ = 25 mm) as references. Some of the compounds that showed activity against *C. albicans* ATCC 10231 were further tested against *C. krusei* ATCC 6285, *C. glabrata* ATCC, and *C. tropicalis* ATCC [92].

Eleven compounds showed antibacterial activity, more potent against Gram-positive bacteria than against Gram-negative bacteria. Compounds **7b**, **8b**-**d**, and **8h** showed inferior activity against *S. aureus* (IZs = 12–16 mm) compared to gentamycin (IZ = 19 mm), while thiadiazolines **8a**–**d** and **8f**–**j** showed superior activity against *E. faecalis* (IZs = 12–18 mm) compared to the reference drug (IZ = 8 mm). Thiadiazolines **8a**, **8c**, **8d**, **8i**, and **8j** were less active against *L. monocytogenes* (IZs = 10–16 mm) than gentamycin (IZ = 18 mm), but showed comparable activity against *B. cereus* (IZs = 16–18 mm) [92].

Gram-negative strains were less susceptible to the synthesized compounds. Compound **8i** was the most potent against *E. coli* (IZ = 20 mm), but the activity was inferior to that of gentamycin (IZ = 22 mm), while thiosemicarbazone **7b** was the only compound from this class active against *S. typhimurium* (IZ = 10 mm) [92].

Concerning the antifungal activity, twelve compounds showed comparable activity to fluconazole against *C. albicans* (IZs = 14–28 mm). The most active thiosemicarbazone derivative **7a** showed better activity against *C. albicans* and *C. glabrata* (IZ = 28 mm) than fluconazole [92].

Based on the obtained results, thiadiazoline compounds possessed better antibacterial activity than thiosemicarbazones, which was presumed by our research group because cyclizing thiosemicarbazones into thiadiazolines enhanced the antibacterial activity. Most of the thiadiazolines were highly selective against Gram-positive bacteria, while their lack of efficacy against Gram-negative bacteria may be attributed to their poor ability to penetrate the additional outer membrane layer.

Structure-activity relationships (SAR) studies showed that the introduction of halogen atoms on the phenyl ring from the fifth position of the thiadiazoline enhanced the antibacterial activity (**8c**, **8d**, and **8g**), whereas compounds with other substituents (hydroxy, methoxy, and nitro) were less active. Interestingly, the double substitution of the same phenyl ring also increased the overall antimicrobial activity. Therefore, an increase in the lipophilicity of the compound may result in better permeability through the outer membrane layer, expanding the activity of Gram-negative bacteria. Compounds with an acetylamino group in the second position of the thiadiazoline (**8a**–**e** and **8i**–**j**) were more potent antibacterial agents than those with an *N*-R-substituted-acetylamino group. However, triple substitution of the amino group increased the antifungal activity. Finally, the introduction of a thiazole ring in the fifth position of the thiadiazoline ring had no major influence on the antibacterial activity but abolished the antifungal activity (Figure 8) [92].

In conclusion, the results of the studies indicated that oxadiazolines are inactive or have reduced antimicrobial activity. The replacement of the oxadiazoline ring with a thiadiazoline ring improved the antibacterial activity, but it was still inferior to the reference antibacterials. Comparing the results by Gram stain, the activity was superior on Gram-positive bacteria compared to Gram-negative. 

In terms of antifungal activity, most thiadiazolines showed good activity against *C. albicans*, with some cases reporting similar activity to fluconazole (Figure 8). The antifungal activity of the tested thiazolyl-oxadiazolines was dependent on their physicochemical characteristics; the reduction of the polar character caused an increase in their antifungal potential.

### 2.3. 1,3,4-Thiadiazolyl-Thioethers and Schiff Bases

1,3,4-Thiadiazole is a versatile framework for drug design since it can be a replacement for the thiazole heterocycle through bioisosteric substitution. Thus, similarly to thiazoles, the thiadiazole heterocycle can be found in synthetic compounds with a large variety of biological activities, including antimicrobial, diuretic, antileishmanial, antiulcer, anti-inflammatory, antioxidant, anticonvulsant, antidepressant, and anticancer activities [93].

The azomethine group and the thioethers are prominent functions in the structures of various antimicrobial compounds and therefore were chosen as structural elements in the thiadiazole derivatives reported by our research group [93].

Starting from 2-amino-5-mercapto-1,3,4-thiadiazole, the thioethers were synthesized through the nucleophilic substitution of the thiol group in basic conditions with various alkylating agents. The alkylating agents included α-bromoarylethanones (**9a**–**f**), α-haloalkylethanones (**10a**–**b**), and halogenated compounds (**11a**–**e**). The Schiff bases (**12a**–**d**, **13a**–**d**, and **14a**–**b**) were obtained following the condensation of the amine group with various aromatic aldehydes in acidic conditions (Figure 3) [93].

The obtained compounds were assayed for their antimicrobial activity against Gram-positive bacteria (*L. monocytogenes* ATCC 35152, *S. aureus* ATCC 25923, and *B. cereus* ATCC 13061), Gram-negative bacteria (*S. typhimurium* ATCC 13311 and *E. coli* ATCC 25922), and *C. albicans* ATCC 90028 fungal strain, using ciprofloxacin and fluconazole as references. The activity was quantified using the disk diffusion method as inhibition zone diameters [93].

Based on the results, the thioethers showed moderate antibacterial activity but good antifungal activity. Compounds **9a**–**c** were the most active against *S. typhimurium* (IZ = 18 mm) and *C. albicans*, but inferior to the used references (IZ = 26 mm for ciprofloxacin and IZ = 28 mm for fluconazole) [93].

The overall antimicrobial activity was better for the Schiff bases, especially for compounds **12c**–**d**, **13a**–**d**, and **14a**–**b**, with a notably superior activity against all tested strains (IZs = 16–30 mm), except for *S. typhimurium*, compared to ciprofloxacin and fluconazole. Compounds **12c** and **13a** showed equal activity to ciprofloxacin against *S. typhimurium* [93].

Structure-activity relationships showed that the antifungal activity and antibacterial activity against *S. typhimurium* increased in the thioether series following the substitution with an 1-arylethanone rest (**9a**–**f**), compared to the 1-alkylethanone (**10a**–**b**) and aryl-/alkyl-substituted compounds (**11a**–**e**, Figure 9). Switching the thioether with an imine linker improved the overall antimicrobial activity of the compounds. This was observed even for the chromenyl-substituted compounds (**13a**–**d**), which were the most active. This was opposite to the previous observation (Figure 8), which may be due to the different main heterocycle [93].

### 2.4. Alkylidene-Hydrazinyl-Thiazoles

Thiazoles and their derivatives are associated with various biological activities. Additionally, compounds containing the azomethine group (CH=N-) have gained importance because of the physiological and pharmacological activities associated with them, mainly antibacterial and antifungal properties [94,95,96,97]. Also, it has been reported that the introduction of a hydrazone group in the second position of the thiazole ring enhances the antimicrobial activity [98]. 

Hydrazones are versatile chemical entities that could act as intermediates for the development of novel bioactive compounds [99]. The literature studies on hydrazone compounds have shown that they possess diverse pharmacological properties such as anticonvulsant, antimycobacterial, antidepressant, anticancer, analgesic, anti-inflammatory, antiviral, antiplatelet, antimalarial, antimicrobial, cardioprotective, vasodilatory, anti-HIV, anthelmintic, antidiabetic, antiprotozoal, antitrypanosomal, and antischistosomal [100].

Starting from in-house-obtained thiosemicarbazones, our research group synthesized some new heterocyclic compounds with the thiazole nucleus linked to the aryl or hetaryl rings by a hydrazone fragment [101,102,103].

The thiosemicarbazones used in the synthesis of the final series of compounds were obtained according to the previously described methodology by the condensation of aliphatic, aromatic, or heteroaromatic aldehydes or ketones with thiosemicarbazide in absolute ethanol and in the presence of catalytic amounts of concentrated sulfuric acid [91,92,101,104,105]. The obtained thiosemicarbazones were used as a new thioamide component in a Hantzsch condensation in order to obtain the thiazole structures. For this purpose, the thiosemicarbazones were subjected to a reaction with chlorinated or brominated halocarbonyl compounds (Figure 4).

Some of the synthesized compounds, chromenyl-methylene-hydrazinyl-thiazoles **15a**–**b** and cycloalkyliden-hydrazinyl-thiazoles **16a**–**j** (Figure 10), were screened for their antimicrobial activity against Gram-positive *S. aureus* (ATCC 29213) and *B. subtilis* (ATCC 60511), Gram-negative *E. coli* (ATCC 25922) and *P. aeruginosa* (ATCC 10145) bacterial strains, and *C. albicans* (ATCC 10231) fungal strains [102,106].

The antimicrobial activity was evaluated by the agar diffusion assay and minimal inhibitory concentration (MIC) determination. Ciprofloxacin and fluconazole were employed as positive controls.

Among the assayed compounds, only two cyclohexylidene-hydrazinyl-thiazoles **16i**–**j** showed activity against *E. coli* (IZ = 20 mm, MIC = 6.25 µg/mL). The activity was two-fold lower than ciprofloxacin (IZ = 20 mm, MIC = 3.12 µg/mL [102]).

The cycloalkylidene derivatives showed good antifungal activity against *C. albicans* (IZs = 20–40 mm, MICs = 3.12–6.25 µg/mL), similar to fluconazole (IZ = 25 mm, MIC = 3.12 µg/mL). Based on the obtained results, the antifungal activity was favorably influenced by the size increase of the cycloalkylidene ring and by the absence of some polar substituents [102].

The activity was similar when the thiazole ring was either unsubstituted or substituted with hydrophobic residues. The obtained results indicated that the chromone ring bound by a methylene-hydrazine linker to the thiazole ring was not favorable for the overall antimicrobial activity, with compounds **15a**–**b** being inactive [102]. The same observation was also depicted in the development of the in-house antimicrobial oxadiazoline compounds.

In conclusion, the cyclohexylidene-hydrazinyl-thiazoles **16i** and **16j** demonstrated good inhibitory activity against *E. coli*, while the cyclopentylidene-hydrazinyl-thiazoles **16a**–**d** and cyclohexylidene-hydrazinyl-thiazoles **16f**–**j** showed excellent activity against *C. albicans*.

### 2.5. Alkylidene- and Arylidene-Hydrazinyl-Thiazolin-4-ones

Thiazolin-4-one derivatives are known to exhibit diverse biological activities such as antimicrobial, antidiarrheal, anticonvulsant, antidiabetic, antihistaminic, anticancer, anti-HIV, anti-inflammatory, and antiplatelet activating factor [107,108,109,110,111]. Therefore, the possibility of derivatizing some thiosemicarbazones into thiazolin-4-ones was exploited, considering the interesting possibility of accommodating the thiazolin-4 one ring and other moieties (thiazole and chromone) in a single molecular framework with the help of an alkylidene-hydrazine linker [101,106].

The general procedure for obtaining the thiazolin-4-one ring consisted of the treatment of different thiosemicarbazones with chloroacetic acid or ethyl-chloroacetate, refluxing ethanol, and in the presence of anhydrous sodium acetate. The used thiosemicarbazones were obtained by the condensation of thiosemicarbazide with some aromatic and heteroaromatic aldehydes or with some hetaryl-methyl-ketones. Furthermore, the presence of the active methylene group in the fifth position of the thiazolin-4-one ring allowed the possibility of Knoevenagel condensations with various aromatic aldehydes. The reaction was carried out by reflux in acetic acid with anhydrous sodium acetate as the basic catalyst in order to yield the corresponding 5-arylidene derivatives (Figure 5) [101,106].

The 2-(aryl-methylene-hydrazinyl)-thiazolin-4-ones **19a**–**h** and **20a**–**f** were tested for their antimicrobial potential. In order to determine the influence of the derivatization in the fifth position of the thiazolin-4-one ring, the unmodulated compounds **17** and **18a**–**c** were included in this study (Figure 11) [106].

The compounds were tested against Gram-positive bacteria, particularly *S. aureus* and *B. subtilis*; Gram-negative bacteria, namely *E. coli* and *P. aeruginosa*; and *C. albicans* [106].

None of the tested compounds inhibited the growth of *P. aeruginosa* and *B. subtilis*. The replacement of the phenyl-thiazolyl moiety from compound **17** with an aryl moiety (**18a**–**c**) led to the obtaining of molecules with weak action against *E. coli* (IZs = 20–25 mm and MIC = 6.25 µg/mL), compared to ciprofloxacin (IZ = 25 mm and MIC = 1.56 µg/mL). The derivatization of the fifth position was not favorable for most of the tested compounds, as there was a decrease or disappearance of the activity [106].

Regarding the antifungal activity against *C. albicans*, the best results were obtained for the unsubstituted thiazolin-4-ones **17** and **18c** (IZs = 35–42 mm, MIC = 6.25 µg/mL), but the activity was inferior to fluconazole (IZ = 25 mm and MIC = 1.56 µg/mL). Also, the derivatization with a chromone moiety was not favorable [102,106], as previously observed in the oxadiazoline and thiazole series.

It can be concluded that in the case of alkylidene-hydrazone derivatives obtained by using various thiosemicarbazones as raw materials, the antibacterial and antifungal activities were modest. The compounds showed an anti-Candida profile rather than an antibacterial one. The replacement of the hydrazone-thiazole scaffold from compounds **15a**–**b** and **16a**–**j** with a hydrazone-thiazolin-4-one scaffold in compounds **18a**–**c**, **19a**–**h**, and **20a**–**f** led to a decrease in the anti-Candida activity (Figure 12).

Previous studies have shown that the hydrazine-alkylidene linker was not favorable for the antimicrobial activity of the in-house thiazolin-4-one derivatives, as the compounds were inactive or had modest antibacterial or antifungal activities. Starting from these observations, new thiazolin-4-one derivatives were synthesized, in which the hydrazono group from the second position of the thiazolin-4-one ring was replaced by substituted amino groups or various aromatic groups (Figure 12).

The aminoacyl-tRNA synthetases (ligases) are a class of enzymatic targets that play an important role in RNA translation. They are responsible for the precision of ribosomal protein biosynthesis, ensuring that the amino acids are correctly esterified to their corresponding tRNA molecules [112,113]. 

Indolmycin is a natural antibiotic with a structure analog to *L*-tryptophan that can competitively inhibit the bacterial tryptophanyl-tRNA synthetase (TrpRS), showing potent antibacterial activity against *Helicobacter pylori*, *E. coli*, *B. subtilis*, and methicillin-resistant *S. aureus* (MRSA). The TrpRS activates *L*-tryptophan for translation through a tryptophanyl-adenylate intermediate and then links this activated amino acid to the corresponding tRNA molecule (tryptophanyl-tRNA). In the structure of indolmycin, the indol ring is linked by a methylene group to an oxazolin-4-one ring [114,115].

Starting from these observations, it was assumed by our research group that the isosteric replacement of the oxazolin-4-one ring with a thiazolin-4-one ring would maintain the affinity for the targeted enzyme. Thus, our research group synthesized new thiazolin-4-one derivatives, diversely substituted in the second and fifth positions, and docked them against two bacterial tryptophanyl-tRNA ligases, one isolated from *E. coli* (PDB 5V0I) and the other from *S. aureus* (1I6K_P67592) (Figure 13) [112,116].

Based on the computed inhibition constants (Ki), an increase in the interaction with the biological targets for the functionalized compounds in the fifth position of the thiazolin-4-one ring compared to the unsubstituted ones was observed. In total, 16 thiazolin-4-ones presented better binding affinities (BA) to *S. aureus* (BAs from −7.77 to −12.1 kcal/mol) and 12 to *E. coli* (BAs from −8.86 to −12.2 kcal/mol) than indolmycin (BA = −7.67 kcal/mol to *S. aureus* and BA = −8.65 kcal/mol to *E. coli*) [112].

The substitution of the second position of the thiazolin-4-one ring with a bulky substituent enhanced the binding affinities to the targeted enzymes, which led to compounds that better mimicked indolmycin. The best binding energies were obtained for the compounds that had a bulky residue, such as phenylamino or α-naphthylamino in the second position of the thiazolin-4-one ring. The presence of a smaller residue, such as allylamino, led to diminished binding affinities [112].

In order to corroborate the in silico hypothesis, the compounds were initially subjected to in vitro antibacterial screening by using the agar diffusion method against *E. coli* ATCC 25922 and *S. aureus* ATCC 49444. All the synthesized compounds and indolmycin were active, recording moderate antibacterial activity against *E. coli* ATCC 25922 (IZs = 14–22 mm), but lower compared to moxifloxacin (IZ = 27 mm). The same compounds showed modest to good inhibitory activity against *S. aureus* ATCC 49444. The 2-*N*-allyl-substituted compounds **21e**–**h**, the 2-*N*-α-naphthyl-substituted compounds **23a**–**e** (IZs = 14–20 mm), and indolmycin (IZ = 20 mm) exhibited similar or better activity than moxifloxacin (IZ = 18 mm) [112]. 

The 2-(1-naphthylamino)-5-arylidene-thiazolin-4-ones **23a**–**e** were, in general, more active against *S. aureus* ATCC 49444 than the thiazolin-4-ones **21a**–**e** and **22a**–**e**, suggesting that the presence of α-naphthylamine, a bulky fragment, in the second position of the thiazolin-4-one ring was more favorable for the antibacterial activity against Gram-positive strains than the presence of an allylamine (series **21**) or phenylamine (series **22**) fragment. This could have been because of some differences regarding the compounds’ intracellular uptake [112].

Moreover, the fact that compound **21f**, substituted with a 3-bromobenzylidene moiety, was more active than compound **21a**, with a 3-chlorobenzylidene moiety, against both tested bacterial strains suggested that the presence of a bulkier halogen atom, such as bromine, in the third position of the benzene ring enhanced the antibacterial activity. Additionally, the compounds bearing two chlorine atoms were generally more active than the compounds with only one chlorine atom, suggesting that more halogen atoms in the molecule were favorable for the antibacterial activity (Figure 14). This may have been due to an increase in the lipophilicity of the compounds, which enhances the intracellular uptake by the bacteria [112].

Prompted by the results obtained in the antimicrobial screening using the agar diffusion method, minimum inhibitory concentrations and minimum bactericidal concentrations (MBC) were determined employing the broth microdilution method. Analyzing the obtained results against *S. aureus*, 12 compounds exhibited similar or better MIC values (MICs = 0.97–31.25 µg/mL) than moxifloxacin (MIC = 31.25 µg/mL), and 15 compounds presented similar or better MBC values (MBCs = 1.95–62.5 µg/mL) than moxifloxacin (MBC = 62.5 µg/mL). The strain of *E. coli* was less sensitive to the activity of thiazolin-4-one derivatives, which displayed lower MIC (MICs = 7.81–125 µg/mL) and MBC values (MBCs = 15.62–250 µg/mL) than moxifloxacin (MIC = 1.95 µg/mL and MBC = 3.9 µg/mL), in agreement with the IZ diameters [112].

All the synthesized thiazolin-4-one derivatives showed moderate to good antibacterial activities. Overall, the compounds were more active against the Gram-positive bacterial strain than against the Gram-negative bacterial strain used in the antibacterial activity assays. The thiazolin-4-one derivatives **21h** and **23a** displayed the best antibacterial activity against *S. aureus* (MIC = 0.97 µg/mL and MBC = 1.95 µg/mL), similar to indolmycin (MIC = 0.97 µg/mL and MBC = 1.95 µg/mL), and 32-fold better than moxifloxacin (MIC = 31.25 µg/mL and MBC = 62.5 µg/mL). The most active substituted compound against *E. coli* was the 2,4-dichloro-phenyl substituted thiazolin-4-one **23b** (MIC = MBC 15.62 µg/mL), which had better antibacterial activity than indolmycin (MIC = 31.25 µg/mL and MBC = 62.5 µg/mL), but was lower than moxifloxacin (MIC = 1.95 µg/mL and MBC = 3.9 µg/mL). The calculated MBC/MIC ratio suggested bactericidal effects for these compounds (calculated ratio = 1–2; standard bactericidal ratio ≤ 4) [112,117].

Lanosterol 14α-demethylase (or CYP51A1) is a key enzyme in the synthesis of ergosterol, an essential component of the fungal cell membrane and an important target for azole antifungals. The enzyme catalyzes the conversion of lanosterol into demethylated precursors of ergosterol through *C^14^*-demethylation [118].

Virtual screening (VS) is an important tool for the identification of good leads and is one of the first essential steps in the drug discovery process. Therefore, the VS output prioritizes the development of the most promising compounds (drug-like or lead-like molecules) prior to high-throughput screening (HTS) [119].

Considering the numerous data from the literature attesting to the fungal lanosterol 14α-demethylase inhibition ability of antifungal azoles, our research group repurposed the previously presented thiazolin-4-one derivatives as inhibitors of this enzyme. All compounds (Figure 13) were tested against Candida strains.

All designed thiazolin-4-one derivatives and two reference compounds, fluconazole and ketoconazole, were docked against the generated homology model for lanosterol 14α-demethylase, using as a template a validated experimental structure from *Saccharomyces cerevisiae* (PDB ID: 4WMZ) [116]. The results indicated that most of the screened derivatives and fluconazole interact by hydrogen bonds, especially with the polar uncharged side chains of amino acid residues such as asparagine, serine, and threonine. Meanwhile, 10 derivatives (**21b**, **21d**, **22b**–**e**, and **23c**–**e**) and ketoconazole did not interact with their target by hydrogen bonds. Two derivatives (**21e** and **22a**) interacted with the amino acid residues through hydrophobic side chains, and only one compound (**23b**) interacted with histidine. Compounds **22d**, **22e**, and **23a**–**e** were stronger inhibitors than ketoconazole.

The compounds were initially subjected to antifungal screening using the agar diffusion method against *C. albicans* ATCC 10231 fungal strain and fluconazole as a reference. All the synthesized compounds showed moderate to good inhibitory activity against this strain (IZs = 16–22 mm). Of these, compounds **21f**–**g** and **23e** exhibited antifungal activities similar to fluconazole (IZ = 22 mm). The 2-allylamino-5-arylidene-thiazolin-4-ones **21a**–**h** were generally the most active against *C. albicans* ATCC 10231, suggesting that the presence of an allyl substituent at the exocyclic amine linked to the second position of the thiazolin-4-one ring was favorable for the antifungal activity (Figure 14).

Prompted by the results obtained in the preliminary antimicrobial assay, all compounds were tested against four Candida strains, namely *C. albicans* ATCC 10231, *C. albicans* ATCC 18804, *C. krusei* ATCC 6258, and *C. parapsilosis* ATCC 22019, by employing the broth microdilution method and using fluconazole and ketoconazole as references [116].

The antifungal activity against the tested strains showed MIC values ranging from 0.015 µg/mL (compound **23b**) to 31.25 µg/mL and MFC values ranging from 0.015 µg/mL (compound **23b**) to 62.5 µg/mL. Most of the compounds exhibited similar or higher MIC and MFC values compared to fluconazole (MIC = 7.81–15.62 µg/mL and MFC = 15.62–31.25 µg/mL) and ketoconazole (MIC = 3.9–7.81 µg/mL and MFC = 7.81–15.62 µg/mL). Overall, the synthesized thiazolin-4-ones presented good to excellent antifungal activities. The MFC/MIC ratio for all tested compounds ranged from 1 to 4, suggesting that the synthesized derivatives could act as fungicidal agents [116].

In conclusion, 18 in-house thiazolin-4-one compounds were designed as potential anti-Candida agents by docking them in the active site of a homologous model of fungal lanosterol 14α-demethylase, a cytochrome P450-dependent enzyme. The synthesized thiazolin-4-one derivatives were tested for their antifungal properties against several strains of Candida, and all compounds exhibited efficient anti-Candida activity.

### 2.6. Aryl- and Hetaryl-Thiazolidine-2,4-dione Compounds

Thiazolidinedione is an important heterocyclic ring system, serving as a pharmacophore and privileged scaffold in medicinal chemistry since the discovery of its role as an antihyperglycemic agent in the early 1980s and as a specific ligand for PPAR-γ (peroxisome proliferator activated receptor gamma) [120]. The exhaustive research has led to the determination of its vast biological profile with a wide range of therapeutic applications. Thiazolidine-2,4-diones are heterocyclic systems consisting of a five-membered thiazolidine moiety with carbonyl groups in the second and fourth positions.

Variable substitutions can occur in the third and fifth positions, but the substitution in the second position brings out the greatest shift in the structure and the properties of thiazolidinediones. Due to their diverse and flexible nature, thiazolidinediones can exhibit a wide range of pharmacological activities, including antihyperglycemic, antimicrobial, antiviral, antioxidant, aldose reductase inhibitors, anticancer, anti-inflammatory, etc. [121,122,123,124,125,126,127,128].

Based on these observations, another research direction was to obtain new biological potential analogues of thiazolidine-2,4-dione. For the synthesis of the starting material, the literature describes two methods, which are either the use of commercial thiazolidine 2,4-dione (**TZD**) or in-house synthesis by condensing thiourea with monochloroacetic acid [129], which was the chosen option by our research group.

The in-house thiazolidin-2,4-diones were further derivatized at the active methylene from the fifth position through condensation with various aromatic and heteroaromatic aldehydes. These 5-arylidene-thiazolidin-2,4-diones were subsequently subjected to *N*-alkylation reactions, following a two-step process: either the derivatization of the fifth position in the first step, followed by *N*-alkylation (series **24**–**26** and **28**–**33**) or the *N*-alkylation in the first step, followed by condensation of the fifth position (series **27**). The synthesized compounds were *N*-substituted-5-chromenyl-2,4-thiazolidindiones (series **25**–**26**) and *N*-substituted-5-aryl(hetaryl)iden-2,4-thiazolidindiones (series **27**–**33**) (Figure 6) [130,131,132,133,134].

#### 2.6.1. 3,5-Disubstituted-Thiazolidine-2,4-diones

Starting from the growing attention towards thiazolidinediones, related to their antibacterial and antifungal activities, our research group investigated the newly synthesized compounds for their antimicrobial potential in different concentrations (1 mg/mL, 5 mg/mL, and 10 mg/mL) using the diffusion method, compared to gentamycin and fluconazole. The compounds (series **24**–**28**) were tested against four Gram-positive bacterial strains: *L. monocytogenes* ATCC 13932, *S. aureus* ATCC 49444, *B. cereus* ATCC 11778, and *E. faecalis* ATCC 29212; two Gram-negative bacterial strains, *E. coli* ATCC 25922 and *S. typhimurium* ATCC 14028; and one fungal strain, *C. albicans* ATCC 10231.

The investigated chromenyl-thiazolidinediones showed moderate to good antimicrobial activity against the selected strains (IZs = 6–28 mm). In general, Gram-negative bacteria were more susceptible to the new molecules, and the substitution with a short alkyl (**24a**–**c**) or acetamido (**25a**–**d**) group was more favorable (IZs = 6–25 mm). By introducing a second chlorine atom on the chromene moiety (**26a**–**g**), the antibacterial activity against Gram-positive bacteria increased (IZs = 10–28 mm). The results showed that the 3,5-disubstituted compounds were more active than the 3- or 5-monosubstituted thiazolidinediones. The antibacterial activity of the monosubstituted compounds was similar (Figure 15).

Additionally, the activity was not influenced by concentration. Compared to gentamycin (IZs = 8–22 mm), the majority of the compounds showed similar diameters of the inhibition zones, with some of the compounds showing superior activity against *S. aureus* and *L. monocytogenes* (IZs = 20–28 mm for **26f**–**g**) or against *E. coli* and *S. typhimurium* (**24b**–**c** and **25b**, IZs = 22–24 mm) (Figure 15) [130,132,133,134].

The replacement of the chromone moiety with an arylidene (**27a**–**p**) or hetarylidene (**28a**–**h**) residue led to a significant decrease in the antibacterial activity against all tested strains. Similarly, in this case, it was observed a higher activity against the Gram-negative bacteria. It was also observed that the 3,5-disubstituted compounds were more active on all strains, either compared to the unsubstituted compounds or monosubstituted in the fifth position or the third position (Figure 15). All compounds showed inferior activity compared to gentamycin [130,132,133,134].

The antifungal potential was assessed against *C. albicans*, using fluconazole as a reference. All new derivatives displayed moderate to good activity against this strain (IZs = 13–35 mm).

As in the case of the antibacterial activity, it was observed that the chromone derivatives were more active compared to those substituted in the fifth position of the thiazolidine-2,4-dione ring with an arylidene or hetarylidene moiety. The only compound more active than fluconazole (IZ = 25 mm) was compound **28j** (IZ = 35 mm), substituted in the fifth position of the ring with a 2-phenyl-thiazol-4-yl-methylene residue and in the third position with a 4-methyl-2′-methyl-2,4′-bisthiazolyl residue, which may be responsible for the enhancement of the antifungal activity of this compound (Figure 15) [130,132,133,134].

In conclusion, the tested thiazolidinediones showed antibacterial and antifungal activities inferior to the reference compounds. For both activities, the results indicated that the substitution with a chromenyl-methylene residue on the fifth position of the thiazolidinedione ring was favorable. The results also indicated that the 3,5-disubstituted compounds were more active than the monosubstituted ones. The best antibacterial activity was registered against the tested Gram-negative bacterial strains, with no evident relationship observed between the different concentrations of the tested compounds and the obtained diameters of the inhibition zones [130,132,133,134].

A new series of 3,5-disubstitued-thiazolidinediones was developed by our research group as potential dihydropteroate synthase inhibitors. These compounds are substituted on the third position of the thiazolidinedione heterocycle with a para-aminobenzoic acid (PABA) residue, acting as a potential substrate for the enzyme. The general synthetic route consisted of nucleophilic substitution in the third position of the thiazolidinedione ring with an α-haloacetamide derivative of PABA, followed by Knoevenagel condensation in the fifth position of the ring with various aromatic aldehydes (Figure 7) [135].

Dihydropteroate synthase (DHPS) is an important enzyme for microorganisms that catalyzes the reaction between dihydropterin and 4-aminobenzoate, affording dihydropteroate, essential for folate synthesis. The enzyme is targeted by sulfonamides, but the extensive usage has led to mutations in the DHPS gene, resulting in sulfonamide-resistant microorganisms such as *Pneumocystis jirovecii*, *S. pneumoniae*, and *Plasmodium falciparum* [136].

According to the molecular docking of the 3-bromo-benzylidene substituted compound **35h** with DHPS, the carboxyl group of this compound can interact through an ionic bond with the protonated amine of Lys203 residue. Other observed ionic bonds were between the oxygen atoms from the amide bond and the thiazolidinedione ring with the imidazole ring of His241 residue. The aryl-methylidene substituent from the fifth position of the thiazolidinedione ring fits in the hydrophobic pocket of the enzyme, while the carbon atom from the bridge between the heterocycle and the PABA residue acts as a hinge, confining flexibility to the molecule [135].

The compounds **34** and **35a**–**i** were evaluated for their antimicrobial activity against Gram-positive bacteria (*L. monocytogenes* ATCC 13932 and *S. aureus* ATCC 6538P), Gram-negative bacteria (*S. enteritidis* ATCC 13076 and *E. coli* ATCC 10231), and *C. albicans* ATCC 10231 fungal strain, using amoxicillin/clavulanic acid 5:1 (A:CA 5:1) and fluconazole as references. The activity was quantified using the disk diffusion method [135].

Based on the results obtained, all synthesized compounds were less active than the reference drugs (IZs = 20–26 mm for A:CA 5:1 and IZ = 24 mm for fluconazole). The insertion of the aryl-methylidene substituent in the fifth position of the thiazolidinedione ring (**35a**–**i**) was responsible for an enhancement in the overall antimicrobial activity compared to the 3-monosubstituted compound (**34**) [135], which supported the previously made observations about the development of antimicrobial thiazolidinedione drugs (Figure 14).

#### 2.6.2. *N*-Substituted-5-Hydroxyarylidene-Thiazolidine-2,4-diones

Infections caused by invasive and pathogenic fungi, especially in high-risk and immunocompromised patients, represent some of the most life-threatening diseases worldwide. The opportunistic yeasts that belong to the Candida genus are the most common human fungal pathogens. The currently available antifungal azoles are widely used in fungal infections, but they have some systemic toxicity and pharmacokinetic deficiencies. These aspects, corroborated by the risk of drug resistance, led to the necessity for novel antifungal agents [137,138].

Based on these facts, the *N*-substituted-5-hydroxyarylidene-thiazolidinedione analogs **29**–**33** (Figure 6) were initially subjected to screening against *C. albicans* by determining the diameter of the inhibition zone [139,140].

According to the obtained results, it was observed that the antifungal activity increased with the *O*-alkylation of the *O*- and *N*-non-alkylated thiazolidinediones **29a**–**d** (Figure 6). The 3-alkoxylated compounds (series **32**, Figure 6) showed the best anti-Candida activity. In the case of para-substituted compounds (series **31**, Figure 6), the introduction of an additional methoxy group slightly increased the antifungal activity. All tested compounds showed lower inhibition zone diameters than fluconazole. Regarding the nature of the R_3_ substituent (Figure 6), the best activity was obtained for the compounds in which the alkylation was conducted with chloroacetophenone or chloroacetamide (Figure 16) [139,140].

For the most potent compounds, our research group continued with the MIC and MFC determination against *C. albicans* strains (*C. albicans* ATCC 10231 and *C. albicans* ATCC 18804) and against other strains, which are the etiological agents of invasive mycoses, *C. krusei* ATCC 6258 and *C. parapsilosis* ATCC 22019 [139,140].

The most active compounds were **30a** and **30b** (Figure 6), having similar anti-Candida activity compared to fluconazole, used as a positive control. For these compounds, the MIC and MFC values for both *C. albicans* strains and *C. krusei* were 15.62–31.25 µg/mL, identical to fluconazole. On the other hand, *C. parapsilosis* was less susceptible to the tested compounds compared to fluconazole. The intermediate compounds **29a** and **29b** had an inhibition diameter equal to 12 mm and were considered to have low antifungal activity. The substitution of these intermediates increased the antifungal potential.

Based on the SAR studies, the most active compounds were substituted with aromatic rings (**30a** and **31a**). An enhancement of the antifungal activity was assured by the presence of the methoxy group on the aromatic linker (**31a**–**f**) (Figure 16 and Figure 6).

Additionally, the obtained results confirmed the previous observations. The most active compounds proved to be those *O*- and *N*-alkylated with a bulky substituent, particularly phenyl-carbonyl-methyl (**30**–**33a**). Moreover, the presence of an etheric group in the third position (series **32**) was the most favorable for the antifungal activity (Figure 16 and Figure 6).

The antifungal azoles interact with lanosterol 14α-demethylase through a coordinative bond between a lone pair of electrons from a nitrogen atom and the Fe^2+^ metal ion in the active site of the enzyme. The presence of atoms with lone pairs of electrons in the structure of thiazolidinediones was a motivation to perform a virtual docking study on a lanosterol 14α-demethylase isolated from *C. albicans*. The best binding compounds were considered **30**–**33a** and **32e**. The disubstitution with a large residue, such as acetophenone (**a**), increased the enzyme inhibition potential. The introduction of two large aromatic rings led to better inhibition of the fungal enzyme, probably due to possible π-π interactions with aromatic amino acids such as Phe384, His405, or His317 [140].

The compounds with a methoxy group had a better affinity for the enzyme compared to the unsubstituted derivatives. This group was favorable because it could have led to the formation of a polar interaction with the N-H from the Leu380-His381 peptide bond of the enzyme [139].

In conclusion, the tested compounds exhibited moderate to good antifungal activity. Substitution of the arylidene linker proved to be favorable for the antifungal activity against *C. albicans*, especially in the case of large aromatic fragments with carbonyl groups. For better antifungal activity, in the case of 4-*O*-alkylated-thiazolidinediones, the arylidene linker was supposed to be substituted with a methoxy group. In the virtual docking study on lanosterol 14α-demethylase, the link to the active center of the enzyme was different compared to fluconazole. It was observed that the thiazolidinedione ring acted more as a hinge for the substituents in the third and fifth positions of the ring than a pharmacophore.

#### 2.6.3. Piperazin-4-yl-(Acetyl-Thiazolidine-2,4-dione) Norfloxacin Analogues

The most recent attempt of our research group to develop antimicrobial thiazolidinediones was through the modulation of norfloxacin by attaching a thiazolidinedione moiety to the piperazine from the seventh position of norfloxacin [141].

Norfloxacin was first derivatized with chloroacetyl chloride, affording a chloracetamide derivative, which was further alkylated with various 5-monosubstituted thiazolidinediones and the unsubstituted thiazolidinedione (Figure 8) [141].

The novel compounds were docked into the active site of *E. coli* DNA gyrase. Based on the results, an increase in the molecular substitution was associated with an increase in the affinity for the topoisomerase [141].

Compounds **36** and **37a**–**f** were preliminary tested in vitro for their antimicrobial activity against Gram-positive bacteria (*S. aureus* ATCC 6538P, *L. monocytogenes* ATCC 13932, *B. cereus* ATCC 11778, and *E. faecalis* ATCC 29212), Gram-negative bacteria (*E. coli* ATCC 10536, *E. coli* ATCC 25922, and *S. enteritidis* ATCC 13076), and two Candida strains (*C. albicans* ATCC 90028 and *C. parapsilosis* ATCC 22019), using norfloxacin and ketoconazole as references. The preliminary assay was performed using the disk diffusion method [141].

Based on the obtained results, the compounds were more active against Gram-negative bacteria (IZs = 14–32 mm) compared to Gram-positive bacteria (IZs = 8–23 mm). However, the compounds showed inferior activity compared to norfloxacin (IZs = 26–38 mm). Some of the compounds showed antifungal activity (IZs = 8–19 mm) but were inferior to fluconazole (IZs = 20–25 mm) [141].

Following the preliminary results, the MIC was determined only against Gram-negative bacteria, particularly *E. coli* ATCC 25922, *E. coli* ATCC 10536, *S. typhimurium* ATCC 14028, *S. enteritidis* ATCC 13076, and *P. aeruginosa* ATCC 27853. However, the activity was inferior (MIC = 0.25–128 µg/mL) to norfloxacin (MIC = 0.0625–1 µg/mL) in all cases [141].

Compared to the unsubstituted analogue **36**, the substitution with an additional phenyl ring in compounds **37a**–**f** increased the overall antibacterial activity. This observation was supported by the previous attempts of our research group, demonstrating that 3,5-disubstituted thiazolidinediones are more potent compared to the monosubstituted ones [141].

### 2.7. Thiazolyl-1,2,4-Triazole Schiff Bases

As presented previously, a trend in our research group that overlaps the current tendencies in drug design is the clubbing of two or three heterocyclic molecules in order to obtain more potent compounds or with different biological activities [142,143,144]. Schiff bases, which are known as key intermediates in organic synthesis and are common ligands in coordination chemistry, have been shown to exhibit a broad range of biological activities due to the imine group in their structures [145,146,147]. 

The ample evidence reported in the literature on the biological potential of Schiff bases containing thiazole and triazole heterocycles has led our research group to synthesize novel Schiff bases containing both heterocycles.

The general method known as Hantzsch condensation has been used for the synthesis of 4-methyl-2-phenylthiazole carboxylate. The 4-methyl-2-phenylthiazole-5-carbohydrazide was obtained by treating thiazolyl ester **38** with hydrazine hydrate in ethanol and refluxing for 3 h, using a water bath. Then, hydrazide **39** was transformed into potassium 2-(4-methyl-2-phenylthiazole-5-carbonyl)-hydrazine-carbodithioate (**40**), following the treatment with carbon disulfide and potassium hydroxide at room temperature. Finally, the 4-amino-5-(4-methyl-2-phenylthiazole-5-yl)-4*H*-1,2,4-triazole-3-thiol (**41**) was obtained by treating compound **40** with hydrazine hydrate under reflux for two hours.

Schiff bases **42a**–**p** were synthesized by the condensation of compound **41** with various aromatic or heteroaromatic aldehydes in the presence of concentrated sulfuric acid as a catalyst (Figure 9). The compounds were tested for their antifungal activity as potential lanosterol 14α-demethylase inhibitors and for their antibacterial activity as potential DNA gyrase inhibitors.

**Scheme 9 antibiotics-13-00763-sch009:**
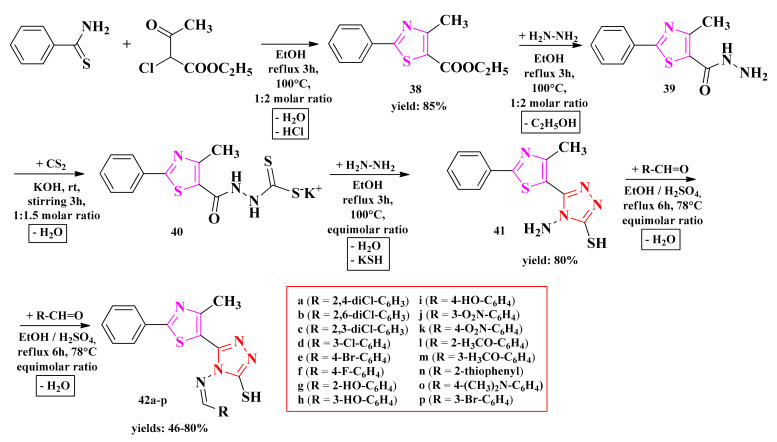
The synthetic pathway for the antimicrobial thiazolyl-1,2,4-triazole Schiff bases [148,149]. Legend: **EtOH**—ethanol; **rt**—room temperature.

Thiazoles, triazoles, and their derivatives play an important role in heterocyclic chemistry due to their biological activity. Fluconazole and voriconazole, which are broad-spectrum antifungals, contain these heterocyclic systems incorporated into their structures. Based on this, the antifungal potential of the synthesized Schiff bases was evaluated in vitro on three Candida strains. The presence of several nitrogen atoms with complexing potential was the initial hypothesis to virtually test these compounds as potential lanosterol 14α-demethylase inhibitors [148,150].

With the aim of elucidating the mechanism of action of the synthesized Schiff bases, molecular docking studies were performed on *S. cerevisiae* lanosterol 14α-demethylase. It was shown that all the thiazolyl-triazole Schiff bases did not covalently interact with the heme from the active site of lanosterol 14α-demethylase as the classical antifungal azoles. However, they interact with the amino acids in the access channel to the active site of the enzyme.

Despite the important role of azoles as pharmacophores for antifungal activity, they also represent a key toxicophore for the hepatotoxicity of the antifungal azoles due to the coordination binding of the nitrogen atoms to the iron atom of the heme. Because the affinity of these Schiff bases for CYP51 was attributed to the non-covalent interaction with the amino acids from the access channel, the presented studies could be useful for the further development of novel antifungal agents that specifically interact with the amino acid residues from the active site, avoiding the toxicity of the classical azoles.

The anti-Candida activity was tested using the disk diffusion method by measuring the diameters of the inhibition zones. The synthesized Schiff bases were screened against *C. albicans* ATCC 90028 fungal strain, using fluconazole as a reference [148,150]. Based on the results, all tested compounds (IZs = 16–20 mm) showed inferior activity to fluconazole (IZ = 25 mm).

Concerning the structure-activity relationships, the compounds substituted on the phenyl ring bound to the azomethine group with para-bromine (**42e**), meta-nitro (**42j**), and para-nitro (**42k**) groups showed the largest inhibition zone diameters (IZ = 20 mm). The other compounds, excepting the 2,4-dichloro (**42a**) substituted compound (IZ = 16 mm), showed the same diameter (IZ = 18 mm) (Figure 17) [148].

For further evaluation of the antifungal potential of compounds **42a**–**p** (Figure 9), their activity was assayed against two less virulent *C. albicans* strains (ATCC 10231 and ATCC 18804) and *C. krusei* for the minimum inhibitory concentrations (MIC) and the minimum fungicidal concentrations (MFC) determination [148]. Based on the obtained results, all tested compounds showed equal or inferior (MICs = 15.62–62.5 µg/mL) activity to fluconazole (MIC = 62.5 µg/mL) and ketoconazole (MIC = 31.25 µg/mL). The Schiff base **42j** (MIC = 15.62 µg/mL) was four-fold more active than fluconazole (MIC = 62.5 µg/mL) and two-fold more active than ketoconazole (MIC = 31.25 µg/mL) on *C. albicans* ATCC 10231. Compounds **42h**, **42j**, and **42n** (MIC = 31.25 µg/mL) were two-fold more active than fluconazole (MIC = 62.5 µg/mL) and equal to ketoconazole (MIC = 31.25 µg/mL). Concerning the activity against *C. krusei*, compounds **42e** and **42j** (MIC = 31.25 µg/mL) were two-fold more active than fluconazole (MIC = 62.5 µg/mL) and equal to ketoconazole (MIC = 31.25 µg/mL) [148].

Finally, we noticed that the thiazolyl-triazole Schiff base **42j** could be considered the most promising antifungal candidate, as it was more active than fluconazole and had similar activity to ketoconazole against all three tested Candida strains (Figure 17). The MFC/MIC ratio for all tested compounds was 2, suggesting that they could act as fungicidal agents [148].

Bacterial DNA gyrases represent important targets in drug discovery, with fluoroquinolones being the only clinically used inhibitors. Their effect is based on the inhibition of the gyrA subunit, perturbing DNA cleavage, and the introduction of negative supercoils into the bacterial DNA. Due to the bacterial resistance to fluoroquinolones and their side effects and limitations, there is a wide interest in searching for novel gyrase inhibitors from different chemical classes, including benzimidazoles, benzoxazoles, benzothiazoles, thiazoles, and triazole derivatives, which bind differently to the biological target [151].

The imine bond in the Schiff bases provides binding possibilities for different nucleophiles and electrophiles, thus inhibiting enzymes or DNA replication. The isosteric replacement of the quinolones’ 3-carboxyl group, essential for gyrase binding, with an aminothiazole fragment or other azoles led to molecules with improved antimicrobial effects and a wider spectrum of activity [152,153]. Therefore, our research group focused on the investigation of the previously mentioned azolyl-Schiff bases against Gram-positive and Gram-negative bacteria.

A molecular docking study was performed on the DNA gyrase subunits (gyrA and gyrB) of *L. monocytogenes*, which are validated drug targets and used in drug design [154]. It was shown that all the Schiff bases were stronger binders than ciprofloxacin to gyrA but weaker binders to gyrB. All compounds make at least three hydrogen bonds between the azomethine nitrogen, the triazole nitrogen atoms, and the amino acid residues from the TOP4c domain of gyrA. The binding pattern of compound **42h** prevents the binding of both ATP and DNA to gyrA, thus preventing the topological transformation of the bacterial DNA [154].

Compounds **42a**–**o** were screened using the disk diffusion method against two Gram-positive (*L. monocytogenes* ATCC 35152 and *S. aureus* ATCC 25923) and two Gram-negative (*S. typhimurium* ATCC 13311 and *E. coli* ATCC 25922) bacterial strains, using ciprofloxacin as a reference [149].

Regarding the activity against the Gram-positive bacteria, *L. monocytogenes* was the most sensible of the tested compounds, with compounds **42a**–**b** and **42i** showing similar effects to ciprofloxacin (IZ = 18 mm and percentage activity index—AI = 100%), while compound **42j** proved to be more active than the reference (IZ = 20 mm and AI = 111.1%). *S. aureus* was moderately inhibited by the new molecules, with the inhibition diameter zones ranging between 12 and 18 mm, respectively, and the AI between 42.8 and 64.2%, compared to ciprofloxacin (IZ = 28 mm and AI = 100%). Compound **42j** showed the best inhibition against *S. aureus* among the synthesized compounds (IZ = 18 mm and AI = 64.2%) [149].

The inhibitory activity against the two Gram-negative bacterial strains was low when compared to ciprofloxacin (IZs = 22–27 mm and AI = 100%), with all derivatives expressing inferior inhibition zones and activity indexes (IZs = 14–18 mm, AI = 51.8–81.8%).

The broth microdilution method was employed for the minimum inhibitory concentration test. All synthesized compounds **42a**–**p** were tested against two Gram-positive bacterial strains (*S. aureus* ATCC 49444 and *L. monocytogenes* ATCC 19115) and two Gram-negative bacterial strains (*P. aeruginosa* ATCC 27853 and *S. typhimurium* ATCC 14028) [149].

Based on the results, 11 compounds (**42a**, **42e**–**g**, **42i**, **42k**–**p**) showed better activity against *L. monocytogenes* (MIC = 1.95 µg/mL) than ciprofloxacin, while the others had a similar activity (MIC = 3.9 µg/mL). On the other hand, *S. aureus* was less sensitive to the activity of the tested compounds (MICs = 15.62–62.5 µg/mL) than ciprofloxacin (MIC = 1.95 µg/mL) [149].

The growth of *P. aeruginosa* was strongly inhibited by most of the compounds, with **42e**–**f** and **42k**–**p** (MIC = 1.95 µg/mL) showing better activity than ciprofloxacin (MIC = 3.9 µg/mL), while compound **42i** had a similar activity. However, the activity against *S. typhimurium* was modest for all tested derivatives (MICs = 31.25–62.5 µg/mL), compared to ciprofloxacin (MIC = 0.97 µg/mL) (Figure 9) [149].

Based on the SAR studies, the bioisosteric substitution with the thiophene heterocycle (**42n**) enhanced the antibacterial activity against *S. aureus*, *L. monocytogenes*, and *P. aeruginosa* (Figure 18). The substitution with chlorine atoms in the second and fourth positions of the phenyl ring (**42a**), along with the substitution with hydroxy groups in the same positions (**42g** and **42i**), was favorable for the antibacterial activity against *L. monocytogenes* (Figure 18). Bromine (**42e** and **42p**), methoxy (**42l** and **42m**), fluorine (**42f**), and dimethylamino (**42o**) substitutions were favorable for the antibacterial activity against *L. monocytogenes* and *P. aeruginosa* (Figure 18 and Figure 9) [149].

The determination of the minimal bactericidal concentration (MBC) confirmed the previously obtained results when the MIC was investigated. The MBC/MIC ratio suggests that the compounds may exert bactericidal activity [151].

In conclusion, the thiazolyl-triazole Schiff bases were investigated for their anti-Candida potential and for their antibacterial potential against Gram-positive and Gram-negative bacterial strains. 

The antifungal activity of the new derivatives was reported for fluconazole and ketoconazole. Compound **42j** was the most promising antifungal candidate, inhibiting the growth of all three Candida strains used and being more potent than fluconazole. The obtained results suggest that the new series bearing thiazole and triazole scaffolds may be considered for further investigation and optimization in designing novel anti-Candida drugs.

In terms of antibacterial activity, the growth inhibitory activity against *E. coli* and *S. typhimurium* was modest. Compounds **42a**–**b** and **42i** displayed the same effects as ciprofloxacin against *L. monocytogenes*, while the inhibition zone diameter for compound **42j** was larger than that of the reference drug. The MIC values for ten of the azolyl-Schiff bases (**42a**, **42e**–**g**, **42i**, and **42k**–**p**) were lower than for ciprofloxacin against *L. monocytogenes*. Most of the compounds strongly inhibited the growth of *P. aeruginosa*. Compounds **42e**–**f** and **42k**–**o** displayed MIC values smaller than the reference, while **42i** showed the same value as ciprofloxacin. The MBC values were in agreement with the MIC values, while the ratio suggested that the tested compounds may express a bactericidal effect [149].

### 2.8. Thiazoles and Bisthiazoles

Since the thiazole was the main framework for our research group in designing novel drugs, especially antimicrobials, there have been many successful attempts along the way to synthesize various thiazole compounds without other heterocycles in their structures, exploiting all three substitution points available on the heterocycle. Small but practical molecules will be presented in the following [155,156,157,158,159,160,161,162,163].

The usual synthetic route for these compounds is through Hantzsch condensation, followed by different reactions in order to modulate the novel substituents grafted on the freshly obtained thiazole heterocycle. Our research group used a large variety of thioamides (**A1**–**8**) and variously substituted α-halocarbonyl derivatives (**B1**–**14**), thus obtaining an impressive number of compounds (Figure 10).

#### 2.8.1. Thiazol-4-yl-1,4-Phenylene-2-Thiazoles

To follow up on our research regarding novel antimicrobial compounds, two new series of compounds (**43a**–**j** and **44a**–**j**) were synthesized, containing two thiazole heterocycles linked in positions 4, respectively 2, through a *p*-phenylene fragment (Figure 19): series **43a**–**j**—2-phenylamino-4-thiazolyl-1,4-phenylene-2-thiazoles, and series **44a**–**j**—2-methyl-4-thiazolyl-1,4-phenylene-2-thiazoles.

The *N*-phenyl-4-(4-thiazol-2-yl)-thiazole-2-amine derivatives (**43a**–**j**) (Figure 19) were evaluated for their antimicrobial activity. The antibacterial activity was tested against *E. faecalis* ATCC 2912, *S. aureus* ATCC 29213, and *S. typhimurium* ATCC 14028, using spectinomycin as a reference. The antifungal activity was tested against *C. albicans* ATCC 10231 and *C. krusei* ATCC 6258, using fluconazole as reference [155].

Based on the obtained results, all compounds showed superior activity against *S. aureus* (MBCs = 62.5–125 µg/mL) compared to the reference (MBC = 250 µg/mL). The activity against *E. faecalis* was superior or equal to the reference (MBC = 250 µg/mL) for all the compounds (MBCs = 62.5–250 µg/mL). Compounds **43d** and **43f** showed equal activity to the reference against *S. typhimurium* (MBC = 62.5 µg/mL) [155].

In terms of antifungal activity, compound **43e** was the only one superior against both tested fungal strains (MFC = 15.62 µg/mL) compared to fluconazole (MFC = 31.25 µg/mL). The rest of the compounds showed equal or inferior activity (MFCs = 31.25–62.5 µg/mL). Based on the MBC(MFC)/MIC ratio, which ranged between 1 and 4, all compounds could act as bactericidal or fungicidal depending on the strain [155].

The in silico studies suggested through molecular docking that 1,4-phenylene-bisthiazoles could primarily engage with lanosterol 14α-demethylase via hydrophobic interactions. None of these chemical compounds were expected to establish polar interactions, either within the active site or with the amino acid residues within the enzyme’s active site. The new compounds **44a**–**j** (Figure 19) were anticipated to function as non-competitive inhibitors of lanosterol 14α-demethylase. This suggested mechanism of action has been reported in the literature as a potential strategy to mitigate fungal resistance to classical azoles [148].

Starting from these observations and taking into consideration the anti-Candida potential of some Schiff bases, thiosemicarbazones, or hydrazones containing the imine group (C=N) [148,164,165], compounds **44g** (R_3_ = -COCH_3_) and **44h** (R_3_ = -OCOC_2_H_5_) were used to obtain the imine derivatives series **45a**–**g** and hydrazone-hydrazide series **46a**–**i** as antifungals with improved activity (Figure 19) [156,157]. The compounds were evaluated for their antifungal activity against pathogenic *Candida sp*. strains (*C. albicans* ATCC 10231, *C. albicans* ATCC 18804, *C. krusei* ATCC 6258, and *C. parapsilosis* ATCC 22019) [155,158,166,167,168].

The tested compounds had MIC values ranging between 3.9 and 31.25 μg/mL and MFC within the interval of 7.81 and 62.5 μg/mL. Compound **45f**, containing a *para*-bromophenyl substituent, exhibited the best antifungal activity, having a potency four times greater than fluconazole against *C. albicans* ATCC 10231 and a potency two times greater against *C. albicans* ATCC 18804, *C. krusei*, and *C. parapsilosis* ATCC 22019 [157].

Compounds **46d** and **46f** demonstrated equivalent anti-Candida activity to fluconazole on the *C. albicans* strains, displaying similar values of MIC = 15.62 μg/mL and MFC = 31.25 μg/mL. Also, compounds **46d** and **46f**–**h** exhibited the same MIC and MFC values as those of the reference drug on *C. krusei* strains. The *C. parapsilosis* strain appeared to exhibit lower susceptibility, with compound **46g** being the only one showing inhibitory activity equivalent to fluconazole (MIC = 7.81 μg/mL and MFC = 15.62 μg/mL) [158].

The anti-Candida activity in the series **46** of hydrazide-hydrazone 1,4-phenylene-bisthiazole derivatives was increased by the presence of the hydroxyl group in the ortho position of the phenyl ring from the hydrazone part (**46d**), compared to the para position (**46e**) (Figure 20). The introduction of a halogen atom in the para-position of the same phenyl ring could improve the anti-Candida activity, as seen in compounds **46f**–**g** (Figure 20). Compound **46h** (2,4-di-Cl-phenylene derivative) showed decreased activity against *C. parapsilosis*, while compound **46i** (thiophen-2-yl derivative) did not improve the anti-Candida activity [158].

In conclusion, multiple series of thiazol-4-yl-1,4-phenylene-2-thiazoles (series **43**–**46**) were designed and evaluated for their antibacterial activity (series **43**) and for their antifungal activity against various *Candida* sp. strains (series **43**–**46**). An initial in silico molecular docking study was conducted to assess the antifungal potential by binding to lanosterol 14α-demethylase, followed by an in silico ADME/Tox analysis, which revealed that these series possess drug-like properties and have a biologically active framework favorable for further improvements [157,158].

#### 2.8.2. 4-(5-Salicylamide)-Thiazoles

According to the literature evidence [169,170,171], salicylamide derivatives are potentially promising antimicrobial drugs. Therefore, starting from this observation, our research group designed two *O*-alkylated series of 4-(5-salicylamide)-thiazoles [155,159].

Both series contain two variation points, the first one in the second position of the thiazole ring (anilino for series **47** and methyl for series **48**) and the second one at the hydroxyl group from the salicylamide moiety. The hydroxyl group of this moiety was alkylated with miscellaneous α-halocarbonyl derivatives and alkyl halides (Figure 21) [155,159].

The series **47a**–**d** was evaluated for the antibacterial activity against *E. faecalis* ATCC 2912, *S. aureus* ATCC 29213, and *S. typhimurium* ATCC 14028, using spectinomycin as a reference, while the antifungal activity was tested against *C. albicans* ATCC 10231 and *C. krusei* ATCC 6258, using fluconazole as a reference [155].

All compounds showed superior activity against *S. aureus* (MIC = 62.5 μg/mL) compared to the reference (MIC = 125 µg/mL). The activity against *E. faecalis* was superior or equal to the reference (MIC = 125 µg/mL) for all the compounds (MICs = 62.5–125 µg/mL). All compounds showed inferior activity to the reference (MIC = 31.25 μg/mL) against *S. typhimurium* (MIC = 62.5 µg/mL) [155].

In terms of antifungal activity, only compound **47a** showed equal potency to fluconazole (MIC = 31.25 μg/mL) against both tested *Candida* sp. strains, while the rest of the compounds were inferior (MIC = 62.5 μg/mL) to the reference [155].

The series **48a**–**o** (Figure 21) was designed as potential lanosterol-14α demethylase inhibitors with antifungal activity, following their docking into the active site of the enzyme. According to the docking results, the most favorable compounds for binding to the enzyme were substituted in R_2_ with an aromatic substituent [159].

The compounds were evaluated in vitro for their antifungal activity against *C. albicans* ATCC 10231, *C. parapsilosis* ATCC 22019, and *C. zeylanoides* ATCC 201082, using fluconazole as a reference [159].

All compounds showed inferior activity (MICs = 62.5–125 μg/mL) against the tested *Candida* sp. strains compared to fluconazole (MICs = 7.81–15.62 μg/mL), except for compound **48n** (containing a para-bromophenyl substituent), which had equal antifungal activity to the reference against *C. albicans* and *C. zeylanoides* (MIC = 15.62 μg/mL) (Figure 21) [159].

#### 2.8.3. 4,5′-Bisthiazoles

Starting from the hypothesis that two thiazole heterocycles directly linked in a scaffold could be potentially favorable for designing novel antimicrobial compounds, as seen in natural compounds (antifungal myxothiazoles and cystothiazoles) and synthetic compounds reported in the literature, our research group synthesized some antimicrobial bisthiazoles [160,172].

Therefore, three series of 4,5′-bisthiazoles variously substituted in the second position with aryl (**49a**–**e**), amino (**50a**–**i**), and hydrazone (**51a**–**c**) substituents were synthesized as potential antimicrobial compounds (Figure 22) [160].

These compounds were evaluated through the disk diffusion method against *S. typhimurium* ATCC 13311, *S. aureus* ATCC 25923, *L. monocytogenes* ATCC 35152, *E. coli* ATCC 25922, *B. cereus* ATCC 13061, and *C. albicans* ATCC 90028, using ciprofloxacin and fluconazole as references [160].

According to the SAR studies of these compounds, the amino substitution of the thiazole heterocycle (**50a**–**i**, Figure 22) was the most favorable for the overall antimicrobial activity compared to the other two series. Compound **50g** showed superior antifungal activity (IZ = 30 mm) to fluconazole (IZ = 28 mm) against *C*. *albicans*. Notable results were also obtained against *B. cereus*, with seven compounds showing superior activity (IZs = 16–26 mm) to ciprofloxacin (IZ = 14 mm). Additionally, compounds **50b** and **50g** (IZs = 20–22 mm) showed superior antibacterial activity to the reference (IZ = 12 mm) against *S. aureus*, where the guanidine (**50b**) and methylamino (**50g**) substitutions enhanced the overall antimicrobial activity (Figure 22) [160].

### 2.9. Thymolyl-Azoles

#### 2.9.1. Thymolyl-Thiazoles

Starting from the previous results, a pharmacophore model that contains the thiazole heterocycle as a centerpiece was built according to the ligand-based pharmacophore hypothesis [158]. 

SAR and molecular docking studies have shown that for proper antifungal activity, this pharmacophore model should contain a heterocycle as a central feature with an available lone pair of electrons acting as a hydrogen bond acceptor (HBA). This heterocycle should be linked to other hydrogen bond donor (HBD) and/or acceptor sites, such as hydrazones or carbonylic moieties, and also to hydrophobic features (HF), such as aromatic or other lipophilic moieties [173].

Thymol (2-isopropyl-5-methylphenol) is a monoterpene phenol mainly found in the essential oils isolated from the Lamiaceae family of plants. The thymol-rich essential oils have been extensively studied for their antimicrobial potential, as it has been shown that thymol is a broad-spectrum antibacterial and antifungal compound [174,175].

Starting from these observations, four novel antimicrobial series (series **52**–**55**) have been synthesized, in which the initial pharmacophore was simplified following the elimination of a thiazole heterocycle.

The hydrophobic feature of this new model was insured by the aromatic moiety linked to the fourth position of the thiazole ring, while in the second position of the same ring, a new fragment was introduced in order to secure the HBA and HBD domains of the molecule. To increase the lipophilicity and the antifungal potency of these series, the newly introduced fragment contains a thymol rest that was connected to the rest of the molecule through an etheric bond (Figure 23) [161].

Starting from the previous pharmacophoric model, two scaffolds were developed: one that kept intact the hydrazone fragment (HBD) linked to thymol by an methylene bridge and supplementary substituted with a phenyl ring as a hydrophobic feature (series **53** and **55**), and the other one as a simplified version that suppressed the acyl-hydrazone fragment and directly linked the thymol moiety in the second position of the thiazole ring through the methylene bond (series **52** and **54**) (Figure 23) [158,161,162].

The general synthetic route of these compounds consists of a Hantzsch condensation, in which the thioamide component was obtained following the *O*-alkylation of thymol with α-halocarbonyl compounds, particularly 2-iodoacetamide (series **52** and **54**) or 2-bromoacetophenone (series **53** and **55**). These intermediates were either treated with Lawesson’s reagent and underwent a thionation reaction or were condensed with thiosemicarbazide. The carbonylic component was represented by variously substituted α-bromoacetophenones (Figure 11) [161,162].

The compounds were subjected to a molecular docking study with the fungal lanosterol 14α-demethylase. The results have shown that the compounds inhibited this enzyme through a non-competitive mechanism [162]. 

In order to confirm the preliminary in silico data, the thymolyl-thiazole derivatives were evaluated for their antifungal potential against *C. albicans* ATCC 10231, *C. parapsilosis* ATCC 22019, and *C. zeylanoides* ATCC 201082 fungal strains using fluconazole as a reference [161,162].

Concerning the antifungal potency in series **52** and **53** (Figure 11), the 2-hydrazinyl-4-phenyl-1,3-thiazole derivatives **53a**–**e** exhibited superior activity to those lacking the *C^2^*-hydrazone linkage **52a**–**e**. The second series of compounds had an increased lipophilicity compared to the series **52**, which was evidenced by the calculated logP values, consequently leading to an increased ability to penetrate the fungal cell membrane. Compared with fluconazole (MIC = 15.62 µg/mL), the best MIC values (MIC = 7.81 µg/mL and 3.9 µg/mL, respectively) were identified for compounds **53a** and **53e** [161].

In the case of compounds **53a**–**e**, some SAR observations could be noticed involving the para-substitution of the *C^4^*-phenyl ring. Compared to hydrophilic and polar substituents (–CN, –NO2, –OH), the presence of a lipophilic, electron-donating substituent (-CH_3_) was linked to improved inhibitory activity on Candida strains. Due to their antifungal effect, hydrazonyl-thiazoles possessing hydrophobic characteristics were further supported by the results obtained from in silico antifungal screening [176,177].

In series **54** and **55** (Figure 11), the most promising compounds were **55a**, **55b**, and **55c**, having a lipophilic para-substituent in the *C^4^* position of the thiazole heterocycle (MIC = 3.9 µg/mL and MFC = 7.8 µg/mL) compared to fluconazole (MIC = 15.62 µg/mL and MFC = 31.24 µg/mL), against *C. albicans* ATCC 10231. The results were four times lower compared to the reference drug. In the case of compound **55d**, the naphthyl substituent decreased the inhibitory activity on Candida strains. Also, the presence of a hydrazone substituent in the *C^2^* position of the thiazole improved the overall antifungal activity [162].

In conclusion, the obtained results regarding the thymolyl-thiazoles suggested that the overall *C^2^*-substitution of the thiazole ring and the presence of a para-phenyl moiety had favorably influenced the antifungal activity. The lipophilic *C^4^*-substituent showed effective anti-Candida activity. Therefore, by keeping the key pharmacophores constant (thiazole and 2-hydrazonylthiazole), the drug design of this study consisted of introducing different moieties at the *C^4^*-position of the thiazole ring.

#### 2.9.2. Thimolyl-Triazoles

Azoles, developed as potent anti-Candida drugs, act by inhibiting lanosterol 14α-demethylase. These compounds are responsible for the cross-over inhibition of CYP51 because of the strong affinity of the nucleophilic nitrogen within the azole heterocycle for the heme Fe^2+^ [178]. 

Because of the increased resistance to the action of the azoles already reported in the literature [179], a particular interest was allocated to the 1,2,4-triazole heterocyclic compounds [180]. 3-*S*-substituted-1,2,4-triazole ring systems are characterized by aromaticity and electron-rich properties. This scaffold is capable of binding lanosterol 14α-demethylase through non-covalent interactions, such as π-π stacking, and covalent coordination of the heme Fe^2+^ by the *N^4^* nitrogen atom of the triazole ring [181]. Moreover, the *S*-linker showed an improvement in some parameters, such as water solubility and lipophilicity, in the drug-like profile [182].

Based on these new observations regarding the 1,2,4-triazole heterocycle and the previous hypothesis regarding the impact on the lipophilicity and antifungal potency of a compound following the insertion of a thymol moiety, our research group has proposed a series of thymolyl-triazoles as antifungal compounds.

The in silico simulation indicates that the designed compounds are potent inhibitors of fungal lanosterol 14α-demethylase, and the results encouraged us to further optimize these compounds, aiming to create innovative non-competitive inhibitors of CYP51 [163].

The series was obtained following the condensation of the hydrazide with various *N*-substituted isothiocyanates, resulting in 1,2,4-triazole-3-yl-mercapto intermediates (**56a**–**c**) that were then *S*-alkylated with different α-halocarbonyl compounds (**57a**–**o**) (Figure 24) [163].

The antifungal activity was evaluated in vitro against *Candida* sp. strains (*C. albicans* ATCC 10231, *C. albicans* ATCC 18804, and *C. krusei* ATCC 6258). The compound **56b** (R_1_ = CH_2_-CH=CH_2_) was the most active molecule, exhibiting the same results as fluconazole (MIC = 15.62 μg/mL and MFC = 31.25 μg/mL) against the *C. albicans* ATCC 10231 and *C. albicans* ATCC 18804 strains. Compounds **57d**, **57i**, and **57n**, having a para-chlorophenyl moiety attached to the 3-mercapto group (Figure 24), showed the strongest antifungal effect against *C. krusei* ATCC 6258 strain [163].

## 3. The Development of In-House Azole Compounds with Antibiofilm Activity

Biofilms are microbial communities composed of bacteria or fungi colonies attached to an abiotic surface and covered by an extracellular matrix. Abiotic surfaces include various medical devices, and, therefore, biofilms represent a threat to human health due to their ability to act as virulence factors and increase antimicrobial resistance [183].

For example, the virulence characteristics of *C. albicans* embrace its noteworthy capacity to create polymicrobial biofilms. This ability enables the rise of complex infections involving both bacteria and fungi, which create significant challenges for treatment due to their heightened resistance to antimicrobial drugs [184]. *C. albicans* offers protection against the bactericidal effect of antibiotics by acting synergistically with bacterial species, for example, methicillin-resistant *S. aureus* (MRSA). Moreover, the fungal biofilm promotes the growth and persistence of anaerobic bacterial strains, such as *Clostridioides difficile* and *Bacteroides fragilis* [185].

One of the strategies to fight biofilm formation and avoid antimicrobial resistance is to facilitate the access of the antimicrobial drug to microorganisms inside the biofilm. Therefore, it is important to develop compounds with antibiofilm potential, even if they have low or no antimicrobial potential.

In the last 10 years, a recent preoccupation in our department has been the testing of azole compounds synthesized as potential antibiofilm agents. With this aim, our research group conducted studies on 2-(3,4,5-trimethoxyphenyl)-thiazoles, 1,4-phenylene-bisthiazoles, thiazolyl-1,3,4-oxadiazolines, *N*-(oxazolylmethyl)-thiazolidindiones, and thiazolidin-2,4-dione norfloxacin analogues [141,186,187,188,189].

### 3.1. 2-(3,4,5-Trimethoxyphenyl)-4-Ar-5-R-Thiazoles

Starting from a series of 3,4,5-trimethoxyphenyl thiazole derivatives previously reported [190,191], it was investigated how the presence of the 3,4,5-trimethoxyphenyl moiety affects both the spectrum and intensity of the antibacterial activity, with a particular focus on inhibiting the microbial biofilm formation followed by the inert substratum colonization (**58a**–**i**, Figure 25) [186]. 

The antibiofilm properties were assessed using the biofilm microtiter method [192,193]. The minimum biofilm eradication concentration (MBEC), expressed in mg/mL, was represented by the lowest concentration of the tested compounds that inhibited biofilm growth [192,193]. A good antibiofilm activity was exhibited by compounds **58a**, **58d**, and **58f** against *B. subtilis* ATCC 6683, with MBEC values of 0.25–0.5 mg/mL. Other compounds that showed good activity were **58g** (MBEC = 0.125 mg/mL, against *E. coli* ATCC 25922), **58a** (MBEC = 0.5 mg/mL, against *C. albicans* 393), and **58e** (MBEC = 0.5 mg/mL, against *K. pneumoniae* IC 13420) [186]. 

### 3.2. Thiazol-4-yl-1,4-Phenylene-2-Thiazoles

Targeting sortase A is widely recognized for the reduction of bacterial virulence through the modulation of various essential processes. These include controlling biofilm formation, regulating host cell entry, evading and suppressing immune responses, and acquiring necessary nutrients [187]. 

A series of in-house 1,4-phenylene-bisthiazoles, substituted in the second position of a thiazole ring with an additional phenyl ring, were designed as potential sortase A inhibitors with antibiofilm potential (**59a**–**g**) (Figure 26).

The structure of compounds **59a**–**g** (Figure 26) was proposed through Bernis-Murcko scaffolding, following the analysis of multiple small molecules known as sortase A inhibitors. The scaffold of these compounds is based on five aromatic rings directly linked, which ensures reduced flexibility, which is necessary for increasing the affinity for the sortase A enzyme [187].

Furthermore, molecular docking studies conducted on compounds **59a**–**g** (Figure 26) have predicted their binding to the active site of sortase A through polar interactions [187].

The antibiofilm activity of the compounds **59a**–**g** was evaluated on Gram-negative bacteria (*E. coli* ATCC 8739 and *P. aeruginosa* ATCC 27853) and Gram-positive bacteria (*E. faecalis* ATCC 29212, *S. aureus* ATCC 6538, *S. aureus* BAA 1026, *S. saprophyticus* ATCC 15305, and *B. subtilis* ATCC 6633), using a microtiter plate method [194]. For the measurement of sortase A inhibition potential, biofilm development was used. Compounds **59a**–**g** exhibited no activity against Gram-negative bacterial biofilm formation. Meanwhile, in the analysis of their effects against Gram-positive bacteria, a prominent result against *E. faecalis* was noticed, with MBEC values ranging between 2–16 μg/mL. Modest activity was observed for compounds **59d**–**g** against *S. aureus* BAA1026 [187].

### 3.3. 2-(3-Pyridyl)-Thiazolyl-1,3,4-Oxadiazolines

The sortase A (SrtA) enzyme from *S. aureus* is involved in the formation of microbial biofilms, as was already described above [187]. Strains of *S. aureus* without the sortase A gene cannot maintain and exhibit the LPxTG (Leu-Pro-any-Thr-Gly) motif in the proteins on the surface of their cells, being deficient in establishing acute infections [195]. 

A series of pyridil-thiazolyl-oxadiazoline compounds (**60a**–**g**) were developed as potential SrtA inhibitors. Molecular docking studies suggested two major ways of potentially interacting with the active site of SrtA: not inhibiting the His153-Cys215-Arg 224 catalytic triad (compounds **60d**, **60f**, and **60g**) and inhibiting the His153-Cys215-Arg 224 catalytic triad (compounds **60a**–**c** and **60e**) [188].

The route of the chemical synthesis involved the condensation reaction of the 4-methyl-2-(pyridin-3-yl)-thiazole-5-carbohydrazide with aromatic or heteroaromatic aldehydes in order to obtain Schiff bases, followed by a cyclization with acetic anhydride to yield the pyridyl-thiazolyl-oxadiazolines (Figure 27) [83,188].

The evaluation of the antibiofilm activity was performed against Gram-positive bacterial strains: *B. subtilis* ATCC 6683, *E. faecium* E5, and *S. aureus* ATCC 6538. The obtained results suggested that compounds **60a**–**g** are inactive in preventing *E. faecium* E5 biofilm formation but offer modest efficacy against *B. subtilis* ATCC 6683 biofilm and express good activity against *S. aureus* ATCC 6538 biofilm formation [188]. 

### 3.4. N-(Oxazolylmethyl)-Thiazolidine-2,4-diones

The agglutinin-like sequence proteins family (Als) are the key molecules in *C. albicans* biofilm growth. Als1 and Als3 have special properties in a particular form of adherence, represented by biofilm formation. Als3 is involved in binding other bacterial strains (*S. gordonii*) as well, being a crucial element of polymicrobial biofilms, and rising *C. albicans* virulence by functioning as an invasin at the epithelial cell level or within the endothelial cells lining the vasculature [196,197,198].

Starting from the aspects presented above, four series of potential Candida biofilm inhibitors were developed, containing different structural components found in various active molecules [199,200]. 

The proposed *N*-(oxazolylmethyl)-thiazolidinedione derivatives were in silico evaluated through molecular docking to elucidate a possible mechanism of action and the analysis of the Als proteins’ structure. A high binding potential against most of the Als surface proteins of *C. albicans*, specifically Als1, Als5, and Als6, was demonstrated. Some compounds, such as **65b,** exhibited good interactions with Als3 [189].

Based on the Blümlein-Lewy reaction, subsequently termed “formamide synthesis” by Bredereck and colleagues [201], the 4-(chloromethyl)-2-phenyloxazoles (intermediates **62a**–**d**) were obtained (Figure 12). The cyclization reaction of an amide with α-haloketone was used to form the oxazole heterocycle. The thiazolidin-2,4-dione intermediates **63a**–**d** were attached to the molecule through alkylation reactions with the intermediate compounds **62a**–**d** [189].

As anticipated during the scaffold design, the results were favorable for 14 of the 16 tested compounds (MBECs = 0.039–0.156 mg/mL), with antibiofilm activity superior to berberine (MBEC = 0.312 mg/mL). The most active compound against *C. albicans* ATCC 10231 biofilm was **65d** (MBEC = 0.039 mg/mL) [189].

### 3.5. Piperazin-4-yl-(Acetyl-Thiazolidine-2,4-dione) Norfloxacin Analogues

The previously mentioned norfloxacin analogues (**37a**–**f**, Figure 8) were additionally evaluated for their antibiofilm potential against *S. aureus* ATCC 25923, *E. faecium* DSM 13950, *E. coli* ATCC 25922, *P. aeruginosa* ATCC 27853, *C. albicans* ATCC 10231, and *C. parapsilosis* ATCC 22019, using berberine as a reference [141].

Based on the results, the compounds showed superior antibiofilm activity against *S. aureus* (MBEC = 4.9–39.0 µg/mL), compared to berberine (MBEC = 78.1 µg/mL). Compounds with an extra halogen atom (**37d**–**f**) were better inhibitors against the *S. aureus* biofilm (MBEC = 4.9–9.8 µg/mL). Superior or equal activities to berberine (MBEC = 625.0 µg/mL) were also reported against *E. coli* and *P. aeruginosa* biofilms [141]. 

Regarding the antibiofilm activity against the fungal strains, compound **37b** showed superior activity (MBEC = 78.1 µg/mL) to berberine against the *C. parapsilosis* biofilm (MBEC = 312.5 µg/mL). Compound **37a** (Figure 8) showed superior activity to berberine against both strains (MBEC = 156.2 µg/mL). The rest of the compounds showed equal or inferior activity to berberine [141].

## 4. Conclusions

The azole heterocycles are versatile moieties for drug design due to their ease of obtaining and extended possibilities to interact with different targets, hence the wide variety of compounds they can be found in.

In our research group, there has been a high interest in capitalizing on these advantages in designing and developing novel antimicrobials for almost five decades. Starting with almost inactive compounds, there was constant progress in enhancing the antimicrobial activity based on SAR studies by swapping up heterocycles or inserting linkers with well-known properties supported by the literature.

The first evaluation method for antimicrobial activity was the disk diffusion method, which today is only used as a preliminary method to select promising antimicrobial compounds. In the following years, the assay of antimicrobial potential also evolved, and the newer compounds were tested using the minimum inhibitory concentration determination as well as the minimum biofilm eradication concentration.

Finally, in order to increase efficiency and decrease time spent on research and development, in silico methods were employed, particularly ADMET, molecular docking studies, and structural analysis of some proteins (e.g., AIs).

For the future, our research group plans to continue the tradition of developing antimicrobial compounds using novel scaffolds and applying the observations highlighted in this paper over the years. Currently, novel scaffolds used are polyphenols and the coumarin heterocycle, while interest is shifting to developing hybrid compounds containing two or three heterocycles in the same structure.

An overview of the research activity conducted by our research group allowed us to observe an evolution in the methodology used (from inhibition zone diameters to minimal inhibitory concentrations and antibiofilm potential determination) correlated with the design of azole compounds based on results obtained from molecular modeling. In the context of the worrying increase in microbial resistance, our future efforts will be directed toward the development of novel antibacterial and antifungal compounds with well-defined mechanisms of action. Moreover, the characterization of the compounds will not be limited to pharmacodynamic properties but will also include pharmacokinetic and toxicological properties and QSAR studies. The outcome of our research is the expansion of currently available compound libraries, which already contain a significant number of compounds with antimicrobial potential.

## Data Availability

Not applicable.

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
