# Peer review of "An Insight into Rational Drug Design: The Development of In-House Azole Compounds with Antimicrobial Activity"

_antibiotics, 2024, doi:10.3390/antibiotics13080763_

Round 1

Reviewer 1 Report

Comments and Suggestions for Authors

In the submitted manuscript, Pele and colleagues present a comprehensive review on the development of Azole compounds with antimicrobial activity. Antimicrobial resistance is the key driving force in the development of drug-resistant pathogens. It affects millions of people worldwide. In recent years, the development of antimicrobial agents has gained significant attention from medicinal chemists. In this review, the authors have nicely summarized the recent development in the discovery of therapeutically active thiazole derivatives designed by the research group from the departments of Pharmaceutical and Therapeutical Chemistry in Cluj-Napoca. This is an interesting review and this reviewer supports the publication after addressing the following issues:

1) In the introduction part, structure of the current FDA-approved azole drugs with antimicrobial activity should be included in a figure.

2) Please comment on the significance, impact, and innovation of the antimicrobial drugs.

3) If at all possible, the mechanism of action of various antimicrobials should be included in a separate figure for a better understanding of the readers.

4) Please comment on the physiochemical and pharmacokinetic properties of some of the azole drugs if available.

5) The authors are encouraged to cite some of the relevant literature reports on the development of Azole compounds with antimicrobial activity, such as those found in

(i) Eur. J. Med. Chem. 2023, 259, 115689. (ii) Eur. J. Med. Chem. 2024, 268, 116221. (iii) Eur. J. Med. Chem. Reports 2024, 10, 100120. (iv) Molecules 2021, 26, 3166. (v) Eur. J. Med. Chem. 2023, 250, 115172.

6) Certain parts of the text should be rephrased. For instance:

(i) Line 151-153: “Overall, the oxadiazoline compounds showed a modest antimicrobial activity or no activity at all against the tested bacteria and fungi. These observations led to the obtaining of some compounds in which the oxygen heteroatom in oxadiazolines was replaced by a sulfur heteroatom” I can understand what the authors are trying to say but more specific phrasing is necessary.

(ii) Line 690-692: “The introduction of two large aromatic rings let to probable π-π interactions with aromatic amino acids such as Phe384, His405, or His317, which led to a better inhibition of the fungal enzyme” this should be rephrased.

7) Some minor corrections:

(i) Scheme 7, line 615, “ET3N” should be corrected to “Et3N”

(ii) Scheme 7, “RT” should be changed to “rt”.

(iii) Figure 19, for compounds 45f and 45g, the abbreviation for “para” substitution should be in italics.

(iv) Line 952, the word “para” should be italicized.

(v) Scheme 11, “r.t.” should be changed to “rt”.

Reviewer 2 Report

Comments and Suggestions for Authors

Raluca Pele et al has chosen an interesting and important topic for this review. The manuscript provides a comprehensive overview of the research group's significant contributions to antimicrobial drug development, particularly through the use of azolic heterocycles such as thiazole and other five membered hetero-cycles. Overall, the manuscript is well structured and provides valuable insights into the development of innovative antimicrobial agents, warranting its acceptance for publication after addressing a major comment.

1. A major comment that author may consider to revisit the schemes and write standard reaction condition for better understanding, throughout the manuscript

2. Abstract and conclusion may be re-visited.

Comments on the Quality of English Language

NA

Reviewer 3 Report

Comments and Suggestions for Authors

In the manuscript titled "An Insight into Rational Drug Design: The Development of In-House Azole Compounds with Antimicrobial Activity," the authors review various azole compounds and their applications in drug design. The manuscript, however, does not reflect the authors' perspective on the discussed information, is not detailed, and contains grammatical errors.

  • Schemes are not properly drawn. It is advisable to include functional groups as R, Ar, and Het in the schemes. Additionally, the synthesis conditions in the schemes are not uniform or adequately detailed.
  • Correlating the schemes and figures would make the manuscript easier to understand, such as connecting Scheme 1 with Figures 2, 3, and 4.
  • In the Introduction, while the authors mention that scaffolds have various activities, it would be beneficial to add marketed drugs for context.
  • There are two different azole scaffolds within the same molecules, such as in Figure 1(d), which includes thiazole. It is unclear which scaffold is responsible for the activity.
  • The authors state that the synthesized compounds have great potential for various activities but do not specify these values. A proper structure-activity relationship (SAR) analysis will be provided, highlighting the best and lowest activity values with the corresponding functional groups.
  • It would be beneficial to add the rationale behind these activities.
  • A critical viewpoint on the current state of research and directions for future developments should be added and discussed in the manuscript. This is crucial for manuscripts of this kind.
Comments on the Quality of English Language

Minor grammatical errors

Reviewer 4 Report

Comments and Suggestions for Authors

The review manuscript " An Insight in the Rational Drug Design: The Development of In–House Azole Compounds with Antimicrobial Activity" by Ungureanu et al. systematically reviews and summarizes the discovery of azole antimicrobial compounds by the research group from the departments of Pharmaceutical and Therapeutical Chemistry in Cluj-Napoca during the last more than two decades. The study summarizes the synthetic strategies, antimicrobial biological activities and structure-activity relationship studies of various azoles series compounds.

However, as a reviewer of this paper, I have the following concerns:

1. The results of antibacterial activity of each series of compounds are listed throughout the paper, but it lacks the necessary organization and summary to facilitate readers to read and understand more clearly, and it is suggested to add more tables and heat maps to show the structure-activity relationship.

2. Although it is a review article, the full text is still too long at 48 pages. I think 15 to 20 pages would be more appropriate.

3. The figure illustrations in the article are still rough, so I suggest that they should be re-drawn in a uniform style and the drawing space in each figure should be reduced.

4. The compound design methodology in the review is still as a classical medicinal chemistry derivation method, and lacks the demonstration of new technologies in this field, such as molecular docking and virtual screening. There is a lack of research on the mechanism of action of the compounds. It is worthwhile for the authors to consider how the content of this review can inspire the new generation of medicinal chemistry researchers to design new generation of azole antimicrobial compounds.

5. At the same time, this paper only presents the research of their research group, and lacks comparison with other groups in the research of azole antimicrobial compounds.

Reviewer 5 Report

Comments and Suggestions for Authors

The manuscript entitled "An Insight in the Rational Drug Design: The Development of In–House Azole Compounds with Antimicrobial Activity" is well designed and precisely drafted well sufficient information. the manuscript provides sufficient information for the designing of better anti-microbials. There are few minor concerns on the manuscript as follows:

1. Can include the WHO statistics on AMR to emphasis the global importance.

2. One of the key agenda in the WHO’s One Health approach is to tackle AMR, few statements regarding that can be included in the introduction or a statement in conclusion to widen the for future research directions.

3. It is mentioned that “Eleven compounds showed antibacterial activity, more potent against Gram-positive than on Gram-negative bacteria”, as mentioned in the manuscript it is due to outermembrane difference alone or the drug binding to different drug targets. 

4. If the permeability is the main concern for the activity difference between Gram positive and negative organisms, suggestion to design the effective molecules can be included. For example which regions can be modified based on the SAR activity.

5. Line 230:  The replacement of the oxadiazoline structure with a thiadiazoline was favorable for the antibacterial activity. However, the activity was inferior on all tested bacterial strains compared to the reference antibacterials. The activity was superior on Gram-positive bacteria compared to the Gram-negative. Additionally, most thiadiazolines showed good activity against C. albicans, in some cases similar to fluconazole.  The statements bit confusing, like  “ inferior on all tested bacterial strains compared to the reference antibacterials”,  “the activty was superior on Gram-positive bacteria compared to the Gram-negative”. Can be modified for more clarity. 

6. A ready reckoner table containing the parent molecule and their analogues with their reported anti-microbial activity against the microorganisms can be provided. 

Round 2

Reviewer 2 Report

Comments and Suggestions for Authors

The author has attempted to address the comments made by Reviewer 2 in the manuscript. However, it appears that the author did not fully understand the comments provided. The comments were not properly addressed, and the work requires major revisions before it can be considered for publication. Below are a few specific comments that need to be addressed:

  1. Figure 1: The structures of FDA-approved azoles need to be rechecked for correct stereochemistry and refined for better visual representation.
  2. Figure 3: The R groups in structures 2a-m should be made to resemble those in 1a-m for consistency.
  3. Figure 6: The thiazole structures need to be rechecked and refined to correctly depict R1 and R2.
  4. Schemes 2 and 3: The author should revisit these schemes and provide standard conditions, equivalence, temperature, time, and yield of the products.
  5. Scheme 6: This scheme is somewhat complicated and should be clarified for better readability. Minor corrections are also needed:
    • Specify what "Anh." stands for.
    • Clarify "MW: 100W" and what "20’" means.
    • Include standard conditions, equivalence, temperature, time, and yield of the product.
  6. Schemes 7-12: Standard reaction conditions are not properly mentioned. Each scheme should include equivalence, temperature, time, and yield of the product.
  7. Figure 16: Clarify if R2O and R2 are the same; if not, specify the difference.
  8. Figure 19: For the transformation from 44g to 45a-g, indicate what other reagents were used and specify the functional group changes. Similarly, provide details for 44h.
  9. Figure 23: Decrease the font size for HF, HBA, and HBD for better visibility.
  10. Figure 24: No other reagents or reaction conditions are mentioned. Ensure that all necessary details are provided.

By addressing these points scientifically, the manuscript can be improved significantly.

Comments on the Quality of English Language

NA

Reviewer 3 Report

Comments and Suggestions for Authors

I have reviewed the revisions and responses from the authors. I am satisfied with it. I recommend accepting the manuscript.

Author Response

We would like to thank the reviewer for taking the time to assess our manuscript and for the positive review report about the article “An Insight in the Rational Drug Design: The Development of In–House Azole Compounds with Antimicrobial Activity”.

Reviewer 4 Report

Comments and Suggestions for Authors

Based on the reviewers' suggestions, I appreciate the authors' efforts in making the necessary point-by-point revisions. The SAR figures in the article have been updated and now appear significantly improved compared to the previous version. Regarding the scientific content of the article, I believe it meets the standards required for publication.

However, as I mentioned in my comments on the first draft, I have concerns about the presentation of the content. Specifically, the SAR information is not easily accessible to readers. While I have reservations about the overall presentation of the paper, I defer to the opinions of the other reviewers. If they agree to accept the paper, I will respect their decision.

Comments on the Quality of English Language

Good!

Author Response

We would like to thank the reviewer for taking the time to assess our manuscript and for the suggestions, comments, and recommendations about the article “An Insight in the Rational Drug Design: The Development of In–House Azole Compounds with Antimicrobial Activity”. We have genuinely tried to present the content of this review in a pleasing manner for our potential readers. Without a doubt, we will always be open to offer additional explanations regarding the content of this manuscript, if any potential reader requests them.

Round 3

Reviewer 2 Report

Comments and Suggestions for Authors

Raluca et al., has revised the manuscript and addressed all the comments raised. The manuscript is now accepted for publication.